# Wave-atmospheric modelling, satellite and *in situ* observations in the Southern North Sea: the impact of horizontal resolution and two-way coupling

Kathrin Wahle[1], Joanna Staneva[1], Wolfgang Koch[1], Luciana Fenoglio-Marc[2], Ha T. M. Ho-Hagemann[1], Emil V. Stanev[1]

[1]Institute of Coastal Research, Helmholtz-Zentrum Geesthacht, Germany

[2]Institute of Geodesy and Geoinformation, University of Bonn, Germany

*Correspondence to*: Kathrin.wahle@hzg.de, Phone: +49 4152 871559

## Abstract

The coupling of models is a commonly used approach when addressing the complex interactions between
different components of earth systems. We demonstrate that this approach can result in a reduction of errors in wave forecasting, especially in dynamically complicated coastal ocean areas, such as the southern part of the North Sea – the German Bight. Here, we study the effects of coupling between an atmospheric model (COSMO) and a wind wave model (WAM), which is enabled through an introduction of wave induced drag in the atmospheric model. The numerical simulations use a regional North Sea
coupled wave-atmosphere model as well as a nested-grid high resolution German Bight wave model. Using one atmospheric and two wind wave models in parallel allows for studying the individual and combined effects of the two-way coupling and grid resolution. This approach proved to be particularly important under severe storm conditions because German Bight is a very shallow and dynamically complex coastal area exposed to storm floods. The two-way coupling leads to a reduction of both surface
wind speeds and simulated wave heights. In this study, the sensitivity of atmospheric parameters, such as wind speed and atmospheric pressure to the wave-induced drag, in particular under storm conditions and the impact of two-way coupling on the wave model performance is quantified. Comparisons between data from *in situ* and satellite altimeter observations indicate that two-way coupling improves the wind and wave parameters of the model and justifies its implementation for both operational and climate
simulations.

## 1. Introduction

Wind forcing is considered as one of the largest error sources in wave modelling. In numerical atmospheric models, wind stress is parameterized by the drag coefficient, which is usually considered as spatially uniform over water. In reality, the wind waves extract energy and momentum from the atmosphere as they grow under wind. This effect is greater for young sea states and high wind speed, in comparison to decaying sea and calm atmospheric conditions. Under such conditions, the drag coefficient cannot be considered as independent from the sea-state and uniform in time and space. This dependence needs to be accounted for in coupled atmosphere-wave models. Jenkins *et al.* (2012) demonstrated that the wave field alters the ocean's aerodynamic roughness and air–sea momentum flux, depending on the relationship between the surface wind speed and propagation speed of the wave crests (the wave age). Based on high resolution coupled simulations, Doyle (1995) demonstrated that young ocean waves increase the effective surface roughness, decrease the 10-m wind speeds and modulate the heat and moisture transports between the atmosphere and ocean; and concluded that as a result of this boundary layer modification, the mesoscale structures associated with the cyclone are perturbed. The impact of sea surface roughness has been investigated in studies by Bao *et al.* (2002) and Desjardings *et al.* (2001). As shown by Lionello *et al.* (1998), the two-way wave-atmosphere coupling attenuates the depth of the pressure minimum. In particular, under extreme conditions, non-linearities increase and the intensity of storms can be modified due to feedbacks between waves and the atmosphere. This feedback must be accounted for in coupled models because strong winds cause the drag coefficient of the sea surface to increase, which leads to wind speed reduction and modification of the wind direction (Warner *et al.*, 2010). These effects feed back into the airflow, wind speed and turbulence profile in the boundary layer. Zweers *et al.* (2002) illustrated that the surface drag was overestimated in the atmospheric weather prediction model for high wind speeds, and the intensity of hurricane winds was underestimated in the simulations; they proposed an approach of calibrating the boundary layer parameterization using the one-way coupled model. They tested a parameterization that decreased the surface drag for two hurricanes in the Caribbean and demonstrated that the new drag parameterization leads to much stronger forecasted hurricanes, which were in good agreement with observations. Two-way coupling of wave and atmospheric models is an alternative approach for development of a fully coupled ocean-atmosphere modelling system, as it enhances the description of interactions and exchanges in the atmospheric boundary layer. Accurate modelling of the boundary layer is of utmost importance for long range predictions.

The coupling between atmospheric and wind wave models was first introduced operationally in 1998 by the European Centre for Medium-Range Weather Forecasts (ECMWF). The method, which is based on

the theoretical work of Janssen (1991), has contributed to an improvement of both atmospheric and surface wave forecasts on the global scale. Waves have been recently considered in operational coupled model systems, such as for Meteo-France (Voldoire *et al.* 2012). Breivik *et al.* (2015) incorporated the effects of surface waves onto ocean dynamics via ocean side stress, turbulent kinetic energy due to wave breaking, and the Stokes–Coriolis force in the ECMWF system.

Air-sea interaction is also of great importance in regional climate modelling. Rutgersson *et al.* (2010, 2012) introduced two different parameterisations in a European climate model. One parameterisation uses roughness length and includes only the effect of a growing sea, as proposed by Janssen (1991). The other, uses wave age and introduced the reduction of roughness due to swell. In both cases, these parametrisations had high impact on the long-term averages of atmospheric parameters and demonstrated

that the swell impact on mixing in the boundary layer is significant.

With increasing grid resolution, the impact of coupling on model predictions becomes more important (Janssen *et al.* 2004), thus emphasizing the need for coupling on the regional scales. Spatial and temporal changes in the wave and wave energy propagation are still insufficiently addressed in high-resolution regional atmospheric models. The shallow water terms in the wave equations (depth and current

refraction, bottom friction and wave breaking) play a dominant role near coastal areas, especially during storm events. The wave breaking term prevents unrealistically high waves near the coast. The spray caused by breaking waves modulates the atmosphere boundary layer.

Järvenoja and Tuomi (2002) emphasized the necessity to use wind data with fine temporal discretization in the wave model, part of the regional coupled atmosphere-wave model at the Baltic Sea, to ensure that

the latter reacts physically correctly to rapidly changing winds. No significant difference caused by the coupling, except for the surface wind speeds, has been found in the meteorological model. For the Mediterranean Sea, however, Cavaleri *et al.* (2012) found that reduced wind velocities was compensated by limited deepening of the pressure fields of atmospheric cyclones. Lionello *et al.* (2003) demonstrated the importance of the atmosphere-wave interaction by studying the sea surface roughness feedback on

momentum flux. In addition, a coupled ocean–atmosphere–wave–sediment transport (COAWST) modelling system has been developed for the coastal ocean (Warner et al, 2012, Kummar *et al.*, 2012). For the Balearic Sea, Renault *et al.* (2012) compared atmospheric and oceanic observations and showed that the use of COAWST improved their simulations, especially for storm events. Recently, high resolution, regional and fully coupled models have been further developed, as shown by Katsafados *et al.*

(2016) who used the Mediterranean Sea as an example. They focused on air–sea momentum fluxes in conditions of extremely strong and time-variable winds. They demonstrated that by including the sea-state dependent drag coefficient, effects on wave spectrum and their feedback on momentum flux lead to improved model predictions. For the southern North Sea (the German Bight area), Staneva *et al.* (2016)

demonstrated the role of wave-induced forcing on sea level variability and hydrodynamics, although the

effects of wave-atmosphere interaction processes were not considered.

Model outputs can be validated against *in-situ* and space-based observational data from satellite altimetry. The accuracy of the 1-Hz wave height and wind speed derived from altimetry has been estimated in previous studies by comparison with in situ-data assumed as ground-truth over intervals of few years. Analysis of the differences between altimeter and *in-situ* measurements over longer time intervals

provides an estimation of the accuracy of altimeter data relative to in situ-data assumed as ground-truth. Significant wave height derived satellite altimetry has been compared to wave height measured by several wave-riders in Passaro *et al.* (2015). Fenoglio-Marc *et al.* (2015) considers the complete satellite mission duration to derive an estimation of the accuracy for significant wave height and wind speed. The standard deviation is between 40 cm and 15 cm for conventional altimetry (Passaro *et al.*, 2015) and between 30

and 15 for SAR altimetry (Fenoglio-Marc *et al.*, 2015). Slightly different results are also obtained depending on the retracker methods used for the altimeter data processing. Higher accuracy is found in open sea (e.g. 15 cm at FINO3 platform) then near coast (34 cm at Helgoland).  They showed that the standard deviations depend on the location of both measurements and on the retracking processing used.

Our objective here is to quantify the effects of coupling between the waves and atmosphere model,

especially during extreme storm events. We present intercomparisons between coupled and stand-alone models and validate these models with newly available space-based observational data. In the one-way coupled setup, the wind wave model only receives wind data from the atmospheric model. In the two-way coupled setup, the wind wave model sends the computed sea-surface roughness back to the atmospheric model. Then, we statistically assess the impact of the two-way coupling and validate the two setups

against available *in situ* and remote sensing data. Our novel contribution here is simultaneously running (via a coupler) a regional North Sea coupled wave-atmosphere model together with a nested-grid high resolution German Bight wave model (one atmospheric model and two wind wave models). Using this configuration allows us to study the individual and combined effects of (1) model coupling and (2) grid resolution, especially under severe storm conditions, which is challenging for wave modelling at German

Bight because it is a very shallow and dynamically complex coastal area.

The paper is structured by describing the models used, the coupling and specification of different model setups, period of model integration and available data for validation in Section 2. Validation of the models against satellite and *in situ* measurements are described in Section 3. A discussion on the impact of two-way coupling is provided in Section 4. The paper ends with a summary and an outlook for future research.


## 2. Model description and set-up

### 2.1 The atmospheric model COSMO

The atmospheric model used in the study is the non-hydrostatic regional climate model COSMO-CLM (CCLM) version 4.8 (Rockel *et al.* 2008, Baldauf *et al.* 2011). The model is developed and applied by a number of national weather services affiliated in the Consortium for Small-Scale Modeling (COSMO, see also http://www2.cosmo-model.org/). Its climate model COSMO-CLM (CCLM) is used by the Climate Limited-area Modelling Community (http://www.clm-community.eu/). CCLM is based on primitive thermo-hydrodynamical equations that describe compressible flow in a moist atmosphere. The model equations are formulated in rotated geographical coordinates and a generalized terrain following vertical coordinates. The model uses primitive equations for momentum. The continuity equation is replaced by a prognostic equation for perturbation pressure (i.e., pressure deviation from a reference state representing a time-independent dry atmosphere at rest, which is prescribed as horizontally homogeneous, vertically stratified and in hydrostatic balance).

In our setup, we use a spatial resolution of ~10 km and 40 vertical levels to discretize the area around the North Sea and Baltic Sea (Fig. 1a). Forcing and boundary condition data are taken from the coastDat-2 hindcast database for the North Sea (Geyer, 2014) covering the period 1948-2013 with a spatial resolution of ~24 km (0.22°) and temporal resolution of six hours.

### 2.2 The wave model WAM

WAM Cycle 4.5.4 is an update of the third generation WAM Cycle4 wave model (Komen *et al.* 1994). The basic physics and numeric are maintained in the new release. The source function integration scheme of Hersbach and Janssen (1999) and the reformulated wave model dissipation source function (Bidlot *et al.*, 2005), later reviewed by Bidlot *et al.* (2007) and Janssen (2008) are incorporated. Depth induced wave breaking (Battjes & Janssen, 1978) has been included as an additional source function. Depth and/or current fields can be non-stationary.

The nested-grid setup includes a regional wave model for the North Sea with a spatial resolution of ~5 km (Fig. 1a), and a finer wave model for the German Bight with a resolution of ~900 m (Fig. 1b). These models, which are described by Staneva *et al.* (2016), use a directional resolution of $15^0$ and 30 frequencies, with equidistant relative resolution between 0.04 and 0.66. The boundary values for the North Sea model are taken from the regional model EWAM (European WAM) of the German Weather Service (DWD). The forcing wind data are provided by CCLM (see Section 2.1). The German Bight

wave model uses boundary values of the outer North Sea model and accounts additionally for depth induced wave breaking and depth refraction. The sea state dependent roughness length, according to Janssen (1991), has already been implemented into the WAM-5.4.5, thus for the present study, the model 170 only needed to be adapted for usage with the OASIS3-MCT coupler (see Section 2.3).

*2.3 Coupling of Models*

The WAM and CCLM are coupled via the coupler OASIS3-MCT version 2.0 (Valcke *et al.* 2013). The 175 name OASIS3-MCT is a combination of OASIS3 (the Ocean, Atmosphere, Sea, Ice, and Soil model coupler version 3) from the European Centre for Research and Advanced Training in Scientific Computation (CERFACS) and MCT (the Model Coupling Toolkit) that was developed by Argonne National Laboratory in the USA. Details of properties and usage of the coupler OASIS3 can be found in Valcke (2013). Exchanged fields between the atmospheric and wave models in this study are wind and 180 sea surface roughness length. For the coupling with OASIS3 the modifications in atmospheric model are as in Ho-Hagemann *et al.* (2013), and in the wave model WAM as in Staneva *et al.*, (2016).

We perform one-way and two-way coupled simulations. In the one-way coupled model, the atmospheric model provides wind data for the North Sea wave model via OASIS. This is equivalent to the familiar forcing of a wave model by 10 m wind fields. We will refer to the results of these simulations as 185 COSMO-1wc and WAM-NS-1wc, where '1wc' and 'NS' stand for 'one-way coupled' and 'North Sea', respectively. In the two-way coupled model, the North Sea wave model is forced with winds provided by the atmospheric model and the sea surface roughness lengths are sent back to the atmospheric model, which in return might change the wind speeds. We will refer to the results of these simulations as COSMO-2wc and WAM-NS-2wc, respectively. The coupling time step for all simulations is 3 minutes. 190 This short time step is a great advantage when modelling fast moving storms, in comparison to using stand-alone wave models forced by winds, which are usually available hourly at the most.

The high resolution German Bight wave model, which also runs simultaneously with the CCLM and North Sea WAM, is forced in the two simulations by the CCLM wind and the boundary data provided by the North Sea WAM set-up. Although the German Bight model does not send roughness information to 195 the atmosphere, we will refer to the two differently forced setups as WAM-GB-1wc and WAM-GB-2wc because roughness information is sent to the atmospheric model by WAM-NS-2wc in the second experiment. This study is novel, compared to previous atmosphere-wave coupling research, because with the OASIS coupler we are able to simultaneously run a high resolution coastal model (the German Bight one) that uses winds and lateral forcing provided by the coupled regional atmosphere (COSMO-2wc) and 200 wave (WAM-NS-2wc) models.

### *2.4 Study Period and Data Availability*

The coupled wave-atmosphere model system described in the previous section was used to simulate a
three-month period from October to December 2013. This period was chosen because it includes the time
when the storm Xaver passed over the study area on the 6[th] of December, 2013. This was one of the most
severe storms of the last decade, which originated south of Greenland and rapidly deepened as it moved
eastwards from Iceland over the Norwegian Sea to South-Sweden and further to the Baltic Sea and
Russia. It reached its lowest sea level pressure on the 5[th] of December at 18 UTC over Norway (~970 hPa,
Figs. 2 and 3). At German Bight, the arrival of Xaver coincided in time with high tides. Because of the
high tides and wind gusts of greater than 130 km/h, an extreme weather warning was given to the coastal
areas of north-western Germany (Deutschländer *et al.*, 2013). This storm event was also exceptional
because of its long duration of nearly two days. The surge height reached ~2.5 m, with its maximum at
low water time. During Xaver, two surge maxima were observed (Staneva *et al.*, 2016). Fenoglio-Marc *et
al.* (2015) described the first surge maximum as a wind-induced maximum. the tide gauge records which
was detected by measurements on German Bight. As demonstrated by Staneva *et al.* (2016), the wave-
induced processes contributed to a persistent increase of the surge after the first maximum (with slight
overestimation after the second peak).

In the present study, we perform statistical analysis for the whole period of integration and investigate the
period of extreme storm event Xaver in more detail. The distribution of wind speeds and directions over
the selected period as seen in the waverider data from the *in situ* platform FINO-1 (see Fig. 1b for its
location) is shown in Fig. 2. North-westerly winds are generally dominant, but strong winds (higher than
20 m/s) came from the west and southwest as the Xaver storm moved eastwards. South-easterly and
north-easterly winds are rarely observed at the FINO-1 station.

To validate our experiments, we use wind speed and significant wave height data measured by satellite
altimeters SARAL/AltiKa, Jason-2 and CryoSat-2 over the North Sea (see Fig. 3 with the tracks of the
different satellites over the three-month study period). The first two carry on-board a classical pulse-
limited altimeter that operates in a low resolution mode (LRM), while the CryoSat-2 instrument operates
in an LRM or in Delay Doppler Altimetry (DDA) mode. The CryoSat-2 data used here were extracted
from the RADS database (Scharroo *et al.* 2013), where CryoSat-2 data acquired in DD mode in our
region was processed to generate pseudo-LRM data (PLRM). Accuracy and precision of PLRM data are
slightly lower than LRM and SAR data (Smith and Scharroo, 2015). The altimeter satellites observe along
their ground-track offshore up to a few kilometres from the coast (Fig. 3). Their ground track pattern and
repeat period are different for each of the three missions, as the same location is revisited by each mission

every 27, 10, and 350 days (Chelton *et al.*, 2001). The SARAL/AltiKa data are of special interest in our study because this satellite passed over German Bight during the storm Xaver when the surge was at its maximum (Fenoglio-Marc *et al.* 2015). The *in situ* wave data from four directional waveriders at German Bight are provided by the Federal Maritime and Hydrographic Agency (BSH) (see Fig. 1b for the buoy locations). The wind speed measurements close to the shore of the Island of Sylt, near the Westerland

buoy location, and on the island of Helgoland are provided by the DWD. At station FINO-1 (see Fig. 1b for its location), there were also wind speed measurements available at 50 and 100 m above sea level for the selected period.

**3.   Validation of the results**

### *3.1 Altimeter data*

The long revisiting time of the same location and the global coverage could be considered as intrinsic

characteristics of the satellite altimetry. Therefore, a longer interval of analysis is needed when statistically analyzing the agreement between the altimeter and in situ measurements, collected from waveriders and anemometers.

The tracks during the study period for the three different satellites are illustrated in Fig. 3 Time-series for the period of Xaver storm in regards to in situ wave height and wind speed measurements at the FINO-1

station, and the nearest point observations of the satellite altimeter SARAL/Altika, as it passed over the region at 5:45 on December 6th  (see also Fenoglio-Marc *et al.*, 2015) are also shown in Fig. 3. The wave height and wind speed measured by the SARAL/Altika altimeter (blue symbol) during the Xaver storm are in good agreement with in situ observations.

The differences between the altimeter and in situ measurements over longer time intervals provide an

estimation of the accuracy of the altimeter data relative to the situ-data assumed as ground-truth. Fenoglio-Marc *et al.* (2015) considered both wave height and wind speed derived from DDA, (also called SAR altimetry) located at a distance to coast larger than 10 kilometres and showed that comparison to *in-situ* observations from the same *in-situ* stations network in the German Bight gave standard deviations between 30 and 15 cm for wave height, 1.6 m/s and 1.8 m/s for wind speed. They also found a good

consistency between pseudo-conventional (PLRM) and DDA in the open ocean, with rms differences of 21 cm, and 0.26 m/s for wave height and wind speed respectively. The cross-validation of PLRM and DDA showed for DDA a higher precision in wave height and a lower precision in wind speed (precisions for DDA were 6.6 cm and 5.8 cm/s for wave heights of 2 meter respectively). *In-situ* analysis showed a

higher accuracy for DDA compared to PLRM. As a demonstration, Figure 4 shows the scatterplots for FINO-1 and CryoSat-2 DDA and PLRM measurements. For the wind speed the accuracy of CryoSat-2 DDA and PLRM is similar (std is 1.9 m/s). For the significant wave height, we observe a higher accuracy in DDA than in the standard PLRM retracking (std are 18 and 30 cm, respectively). The accuracy in the significant wave height from PLRM increases (std is 19 cm) when a dedicated retracking procedure is applied (Fenoglio-Marc *et al.*, 2015). Figure 4b shows an underestimation of wind speed of altimetry relative to the in situ data (slope is below 0.8 in all cases).

### 3.2 Altimeter-model comparisons

In this section, we quantify the performance of one-way *versus* two-way coupling by comparing the output of the atmospheric and wave models against remotely sensed data. Table 1 gives the statistics of the differences (bias and standard deviations) between the model and altimeter-derived values of wave height and wind speed over the selected three-month period. The numbers of matched pairs (approximately 7000) of observations and simulations are also given in Table 1 for the different satellites. For all three satellites, the standard deviation in the two-way coupled model is smaller than in the one-way coupled model. Similarly for Jason-2 and SARAL/Altika, the bias in the two-way coupled model is nearly halved compared to the one-way model, due to the reasons explained in the introduction; thus, this finding is the first indication that the model offers a skill improvement. Measured values are lower than the modelled values in the one-way and two-way experiments. In contrast, for Cryosat-2, the opposite is true. In other words, the measured values are higher than the modelled values on average for both the wave height and wind speed. The biases between the CryoSat-2 data and the two-way model simulations (see the red shaded values in Table 1) are larger than the biases between the CryoSat-2 data and the one-way model runs. Fenoglio-Marc *et al.* (2015) also found that the CryoSat-2 derived wave height data overestimate the wave model data from the DWD. However, they found the opposite for the wind speed. i.e. the CryoSat-2 derived wind speed underestimates the COSMO winds from the DWD data. This disagreement is due to the different data that have been used to force the atmospheric models by DWD and this study. This demonstrates again that a determination of wave height from satellite altimetry is particularly challenging for waves smaller than one metre (Passaro *et al.*, 2014).

To perform a spatial comparison between model simulations and the satellite data, we analysed individual tracks over the North Sea, and two of these are shown in Figures 5 and 6. The satellite altimetry observations along the ground-track at the time of the overflight at German Bight last ~38 sec. The selected SARAL/AltiKa passes are very diverse, as one was taken under calm conditions (Fig. 5) and the other pass occurred during the storm Xaver (Fig. 6). Therefore, an opportunity was provided to compare

measured and modelled wave heights and wind speeds along the satellite tracks under different atmospheric and wave conditions. Here, we provide a demonstration only for two tracks, but these tracks offer illustrative comparisons for calm conditions and an extreme storm event. Under calm conditions, differences between the results of the one- and two-way coupling are very small (Fig. 5a). Both models (WAM-NS-1wc and WAM-NS-1wc 2wc) overestimate the measured wave height (red line) over a large part of the track. However, the increased modelled wave height with latitude appears to be consistent with the northward wind speed increase observed by the satellite data and simulated in the two simulations (Fig. 5b). During the storm Xaver, the difference between the wave height in the WAM-NS-1wc and WAM-NS-2wc simulations (Fig. 6a) increases up to 1 m in the southern North Sea. The altimeter-derived quantities fluctuate greatly. However, the two-way coupled-model results are closer to the satellite data, in comparison to the ones in WAM-NS-1wc, except for the latitude ~56 deg. N, where the significant wave height from the satellite measurements has a local peak. The modelled significant wave height (black lines) is much smoother than the satellite observations (red line). This result can be explained by the different post-processing of the significant wave height in the satellite data and by the statistical nature of the wave spectral model. The corresponding wind speed does not grow at this latitude, neither for the measured nor modelled wind speeds. It is noteworthy that the measured peak of the storm is underestimated in both experiments and also shifted northwards by ~2 degrees (Fig. 6a). The modelled wind speed fits well with the altimeter-derived wind speed in the calm situation for both experiments (COSMO-1wc/2wc, Fig. 5b). Northwards of 55 degrees N, the wind speed is higher than 10 m/s, and the wind speed in the two-way coupled experiment (COSMO-2wc, full line) is reduced. During the storm Xaver, the measured wind data fluctuate ~18 m/s, whereas the modelled data show much higher values of ~20 m/s, which reached ~22 m/s at latitudes ~57 and 59 degrees N (Fig. 6b). This finding indicates that the algorithm for retrieving wind speeds is saturated under these extreme conditions. Fenoglio-Marc *et al.* (2015), who had compared the same altimeter data to ERA-Interim, NOAA/GFC and COSMO/EU winds, have suggested that the low wind speeds derived from the altimeter are caused by an overestimation of the atmospheric attenuation of the radar power in Ka-band. Indeed, a larger attenuation correction would result in a too large backscatter coefficient and hence a reduced wind speed. The correction in the SARAL/AltiKa products is larger than the correction based on surface pressure, near-surface temperature, and water vapour content (Lillibridge *et al.*, 2014). The wind speed simulated by COSMO-2wc is lowered up to 1 m/s compared to that of the COSMO-1wc. Similar analyses along all tracks over the study period agree with the two examples demonstrated in Figs. 5 and 6. In general, the measured wind speeds were in slightly better agreement with the two-way coupled model results, which was also demonstrated by statistics presented in Table 1. The track during the time of storm Xaver was the only track taken under such extreme conditions.

### *3.3 Validation against in situ measurements*

Analyses of the temporal variability of the significant wave heights at German Bight under stormy
        conditions allow us to investigate not only the impact of two-way coupling but also the role of the
        horizontal resolution. The comparison between data from two waverider buoys (see for locations Fig. 1b)
        and from the coarse North Sea wave model (WAM-NS-1wc/2wc) and fine German Bight model (WAM-
        GB-1wc/2wc) are exemplified in Fig. 7 during the storm Xaver. Throughout this period, the highest

values of significant wave height are simulated by the WAM-NS-1wc experiment. The lowest values, and
        closest to the observations, are from the WAM-GB-2wc simulations (Fig. 7). At the beginning of
        December, during the calm atmospheric conditions, all model results are similar and fit relatively well
        with the *in situ* measurements. The differences in the wave growth between the different model
        simulations become significant after the storm onset. The peak of the storm, as estimated by the WAM-

NS-1wc simulation, overshoots the measured wave heights by ~3 m at the Helgoland station (water depth
        30 m, Fig. 7a) and by ~4 m at the shallow water of the Westerland station (water depth 13 m, Fig. 7b).
        This peak occurs earlier in all simulations in comparison to the *in situ* measurements, due to the time-shift
        in the wind data. The wave heights predicted by the WAM-GB-2wc  are in better agreement with the
        observations, especially for the Westerland station (Fig 7b, the red-line), in comparison to the other

experiments.

        The influence of spatial resolution on the simulated characteristics can be clearly seen in the time series at
        the deep water buoy at Helgoland. This buoy is located in an area of large gradients in water depth (Fig.
        1b), which explains why the differences of wave height during the storm Xaver reach ~1 to 1.5 m in the
        corresponding North Sea and German Bight simulations (Fig. 7a). This finding identifies the importance

of increasing the horizontal resolution of the models in the coastal areas with complex bathymetry.

        At the shallow Westerland buoy station (Fig. 7b) the differences are additionally enhanced by the depth-
        induced wave breaking in the German Bight model. This can also be seen in the snapshots of wave height
        in the North Sea and German Bight models at the peak of the storm (Fig. 8 a, b). Shoreward of the 15 m
        isobaths, the wave heights drop from 6 to 4 m in the German Bight model. In contrast, for the North Sea

model, the 6 m high waves reach the south-eastern coast. The WAM-NS-1wc underperforms in
        comparison to WAM-NS-2wc at Westerland. This  shows convincingly the importance of two-way
        coupling for the coastal German Bight areas, where the model wind speed is even higher (by ~2 m/s) than
        at Helgoland. We admit that it is difficult to differentiate between effects coming from wave breaking and
        from two-way coupling because both contribute to reducing the wave height by extreme weather

conditions. Wave breaking plays a dominant role in very shallow water, especially during storm events,

by preventing unrealistically high waves near the coast. For the deep waters, the sea surface roughness feedback due to the two-way coupling plays a very important role (Fig. 7a). The importance of the two-way coupling is clearly demonstrated by comparing the WAM-GB-2wc (the blue line) and WAM-GB-1wc (the red line) in Fig. 7. For all stations, the simulated WAM-GB-2wc is reduced, especially during the Xaver peak, and is closer to the measurements. The wind fields in both locations are very similar in the COSMO-1wc/2wc model runs; the peak of the storm is reduced from 26 to 22 m/s. By comparing the model and measured wind speed, it is noticeable that the modelled wind speeds grow too early and too high at all locations at the beginning of the storm (see the bottom patterns in Fig. 7 a,b for the Helgoland and Westerland examples). The storm characteristics are matched well at Helgoland but are slightly underestimated at Westerland. Still, the overall model performance at Westerland is satisfactory, considering the strongly fluctuating wind measurements. Similar behaviours are observed for the Elbe and Fino-1 wave buoy stations.

Additionally, the wind speeds are validated against measured data from FINO-1 in 50 m and 100 m height over the whole modelling period (Table 2). We find better agreement in the two-way coupled run. The bias in wind speed is negative for the one-way coupled setup; thus, the modelled wind speed overestimates the measured wind speed. The bias is significantly reduced due to the lower wind in the two-way coupled model. The root mean squared difference (RMSE) is ~3 m/s in either case, but slightly improved for the full coupled setup.

For a more quantitative validation of the WAM-GB-1wc/2wc results, we use four buoys (see Fig. 1b for their locations) in water depths of 13 to 30 m. Table 3 gives the statistics for significant wave height (*Hs*) over the whole period (there are ~4000 matched pairs). For the four buoys and regardless of the type of coupling, the bias for Hs is slightly negative, i.e.*,* the modelled data over predict the measured values. The simulated significant wave heights are lower and the bias between the measurements and model results are significantly reduced in the WAM-GB-2wc experiment. The standard deviation of the significant wave height of the two-way coupled simulation is similar to that of the one–way coupled simulations. Only for the FINO-1 station, the standard deviation is reduced by ~5% by the two-way coupled model.

**4 Impact of the two-way coupling**

In the following discussion, the impact of coupling is analysed for the North Sea focusing on the spatial patterns under different physical conditions. Three-month averaged significant wave height and wind speed is reduced significantly (Fig. 9) due to the two-way coupling, which results from an extraction of energy and momentum by waves from the atmosphere. The average difference (bias) in wave height (Fig. 9a) is ~20 cm, which is a reduction of ~8% of the three-month mean value (~2.3 m). The RMSE between

the two simulations (Fig. 9b) is ~40 cm in the central North Sea. For the wind speeds, the bias (Fig. 9c) is

~30 cm/s when averaged over the model area, corresponding to a reduction in wind speed of ~3% of the

three-month mean value (~10 m/s). The RMSE (Fig. 9d) between the two-way and one-way coupled

simulations over the whole North Sea area is ~80 cm/s. The spatial patterns in the bias in Fig. 9 can be

explained by the dominant westerly winds (Fig. 2). As the wind comes from land (Great Britain) and

strikes the North Sea, the differences in the wind speed between the two models are larger closer to the

coast because of differences in sea surface roughness. Moving further east, the atmospheric boundary

layer adapts in both cases to the winds over sea, and there is less difference between the one- and two-

way coupled models. For the wave height, differences in bias close to the western coasts and in the

English Channel are small because some fetch is needed for the waves to evolve and the fetch is too short

there.

The differences in the mean sea-level pressure between COSMO-1wc/2wc for the storm Xaver period is

analysed in Fig. 10. The mean sea level pressure at the peak of the storm (Fig. 10a) has values of ~900

hPa over Norway and ~1000 hPa over the North Sea. Compared to the one-way coupled setup, the

pressure increased by ~50-100 Pa in the southeast (Fig. 10b). The slightly decreased pressure in the

remaining part of the model area indicates a shift of the pressure low minimum, confirming the results of

Cavaleri *et al.* (2012), who found similar patterns in the Mediterranean Sea under developing cyclones.

As was noted by Janssen and Viterbo (1996), the timescale of wave impact on the atmospheric circulation

is on the order of five days. However, our model area is too small to observe this impact. More plausible

is that our results are caused by the wave-mean flow interactions in the atmosphere. This effect of wave

coupling on the atmosphere circulation will be analysed more deeply in future experiments.

Another illustration of the influence of the coupling is given by the two time series at the FINO-1 station,

each of about two weeks and taken under very different conditions. One period is in November, which

was rather calm, and contained young and developing wind seas (Fig. 11). The other period was in

December with several storms coming from the North Sea (including Xaver) with higher wave ages (Fig.

12). The differences in significant wave height and wind speed between the one- and two-way coupled

models are mostly positive, i.e., both parameters are reduced in the two-way coupled model. The largest

differences can be observed when the wave age (the ratio of phase velocity at the peak of the wave

spectrum with friction velocity) is well below 20 and occurs before the maximum wave height has been

reached (this can be well seen for Xaver, Fig. 12). Thus, the waves grow slower in the two-way coupled

model. Negative differences seldom occur, only occurring when the wave age increases rapidly (when the

wind speeds go to zero, the wave age goes to infinity).

## 5. Summary and Outlook


We developed a two-way coupled wave-atmosphere model for the North Sea, which includes the possibility of nesting a coastal, high-resolution wave model; the two models perform simultaneously. This analysis was done by using the coupling software OASIS3-MCT, which allows a parallel run of several models on different model grids. By using a coupler, simultaneous simulations of a regional North Sea

coupled wave-atmosphere model together with a nested-grid high resolution German Bight wave model (one atmospheric model and two wind wave models) were performed. This allowed us to study the individual and combined effects of two-way coupling and grid resolution, especially under severe storm conditions, which is challenging for the German Bight, because it is a very shallow and dynamically complex coastal area. The sensitivity of atmospheric parameters such as wind speed and atmospheric

pressure to wave-induced drag, in particular under storm conditions, were quantified. Model intercomparisons gave encouraging results. Overall, the two-way coupled model results were in better agreement with the *in situ* and remotely sensed data of significant wave height and wind speed, in comparison to the one-way coupled model (COSMO drives WAM). New in this paper is the use of satellite altimetry, which provides complementary information to *in-situ* data for the validation of models.

We show that comparisons between the model results and satellite-derived parameters are satisfactory, except for a known degradation of wind speed in storm conditions, which is under investigation. The two-way coupling improved the modelled significant wave heights in the German Bight, which was demonstrated by the validation against *in-situ* observations from four different buoys.

For the storm event Xaver, the impact of the two-way coupling was of highest significance. Wave heights

decreased from ~8 m to ~5 m due to the coupling, which matched buoy measurements very well. The corresponding wind speeds were lowered from ~22 to ~20 m/s. In addition to this extreme event, such large differences between one- and two-way coupled model results were only observed for young seas (wave age well below 20). We also found a slight spatial shift in the minimum of the cyclone mean sea level pressure together with a slight increase of the pressure field from the two-way coupled model runs.

These results may also have been caused by the wave-mean flow interactions in the atmosphere. This will be the subject of subsequent work, where we will study in more depth the consequences of coupling with other atmospheric parameters at sea level and the vertical structure of the planetary boundary layer.

Staneva *et al.* (2016) addressed the impact of coupling between wave and circulation models of German Bight during extreme storm events. They demonstrated that the coupled model results revealed a closer

match with observations than from the stand-alone circulation model, especially during the extreme storm Xaver in December 2013. This study showed that the predicted surge of the coupled model is significantly enhanced during extreme storm events when accounting for wave-current interaction. In our

study, we also demonstrated that for regions such as the German Bight, the role and potential uncertainties of shallow water in the wave model are also of great importance. Shallow water regions with the strongest wave-current interactions contribute largely to the coupled wave-atmosphere dynamics during extreme storm surge events. Depth and current refraction, bottom friction and wave breaking in the wave model play dominant roles in very shallow water. Nevertheless, model resolution is critical where the depth gradients are large. The improved model skills resulting from the new model developments justify further extension of the coupled model system by integrating atmosphere-wave–current interactions to further investigate the effects of coupling, especially on extreme storm events.

**Acknowledgments**

This work has been supported by the Coastal Observing System for Northern and Arctic Seas (COSYNA) and as part of the Horizon2020 CEASELESS project 730030. The authors like to thank Arno Behrens for providing the boundary values for the wave model from his COSYNA results. Beate Geyer extracted boundary values from the coastDat2 database for us. Markus Schultze supported us by setting up the atmospheric model and getting it started. Ha Ho-Hagemann is supported through the German project REKLIM and the Baltic Earth Programme. Luciana Fenoglio is supported by the European Space Agency (ESA) within the Climate Change Initiative (CCI). The authors are grateful for I. Nöhren for assistance with the graphics and BSH for providing the observational data.

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

*Table 1: Bias and standard deviation of validation of wind speed (m/s) and significant wave height (m) of the one- and the two-way coupled models against the available satellite data over the whole period (measured minus modelled).*

| | Significant wave height [m] | | Windspeed [m/s] | |
|---|---|---|---|---|
| | one-way | two-way | one-way | two-way |
| **Saral/AltiKa # 6886** | | | | |
| mean meas. | 2.35 | | 9.76 | |
| bias | -0.27 | -0.12 | -0.64 | -0.33 |
| std. dev. | 0.93 | 0.86 | 3.33 | 3.16 |
| **Jason-2 # 6710** | | | | |
| mean meas. | 2.38 | | 9.62 | |
| bias | -0.29 | -0.15 | -0.73 | -0.40 |
| std. dev. | 1.07 | 1.01 | 3.85 | 3.75 |
| **Cryosat-2 # 7477** | | | | |
| mean meas. | 2.71 | | 10.62 | |
| bias | 0.18 | 0.31 | 0.39 | 0.65 |
| std. dev. | 0.90 | 0.87 | 3.33 | 3.18 |

*Table 2: Wind speed (m/) bias and standard deviation of the one- and the two-*

*upled COSMO model data against the FINO-1 data  over the whole period*

*ured minus modelled).*

| | windspeed [m/s] at 50m | | windspeed [m/s] at 100m | |
|---|---|---|---|---|
| | one-way | two-way | one-way | 610 two-way |
| mean meas. | 11.03 | | 11.85 | |
| bias | -0.67 | -0.41 | -0.23 | 0.01 |
| rmse | 3.26 | 3.17 | 3.33 | 3.22 |

*Table 3: Significant wave height (m) bias and standard deviation of  the one- and two-way coupled WAM*

*German Bight model data against the available buoy  data over  the whole period  (measured minus*

*modelled).*

| bouy name (depth) | FINO-1(30m) | | Elbe (25m) | | Helgoland (30m) | | Sylt (13m) | |
|---|---|---|---|---|---|---|---|---|
| mean meas. hs [m] | 1.95 | | 1.42 | | 1.63 | | 1.45 | |
| | 1-way | 2-way | 1-way | 2-way | 1-way | 2-way | 1-way | 2-way |
| bias hs [m] | -0.14 | -0.03 | -0.07 | -0.01 | -0.13 | -0.03 | -0.15 | -0.05 |
| std. dev. hs [m] | 0.45 | 0.50 | 0.49 | 0.49 | 0.54 | 0.55 | 0.59 | 0.59 |

**Figures**

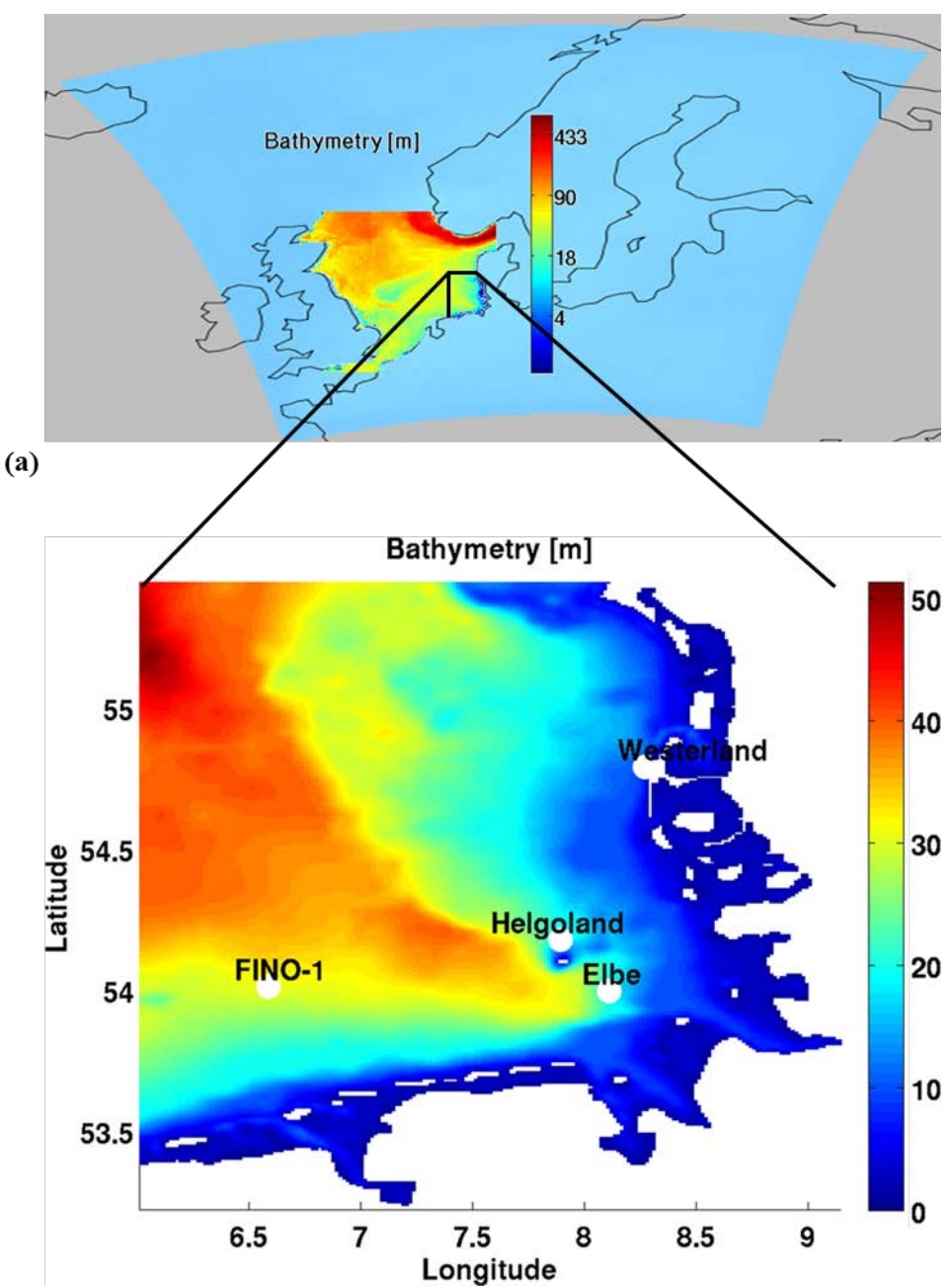

*Figure 1: (a) Bathymetry (m) of the North Sea embedded in the COSMO model area (using a logarithmic scale) and (b) bathymetry (m) of German Bight as used in the WAM model. The positions of four waverider buoys used for the validation is indicated, too.*

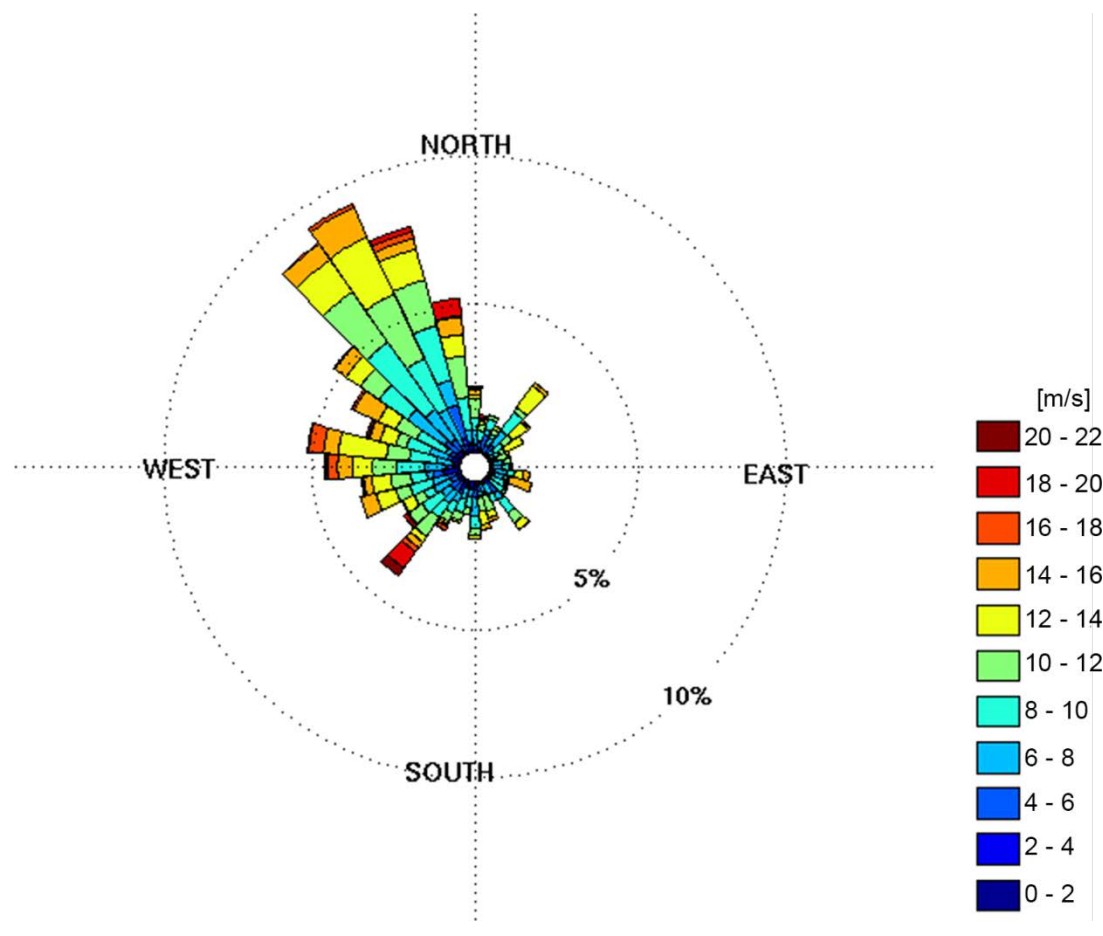


*Figure 2: Distribution of wind speeds in m/s (see color bar) and directions at the FINO-1 waverider buoy for October - December 2013.*


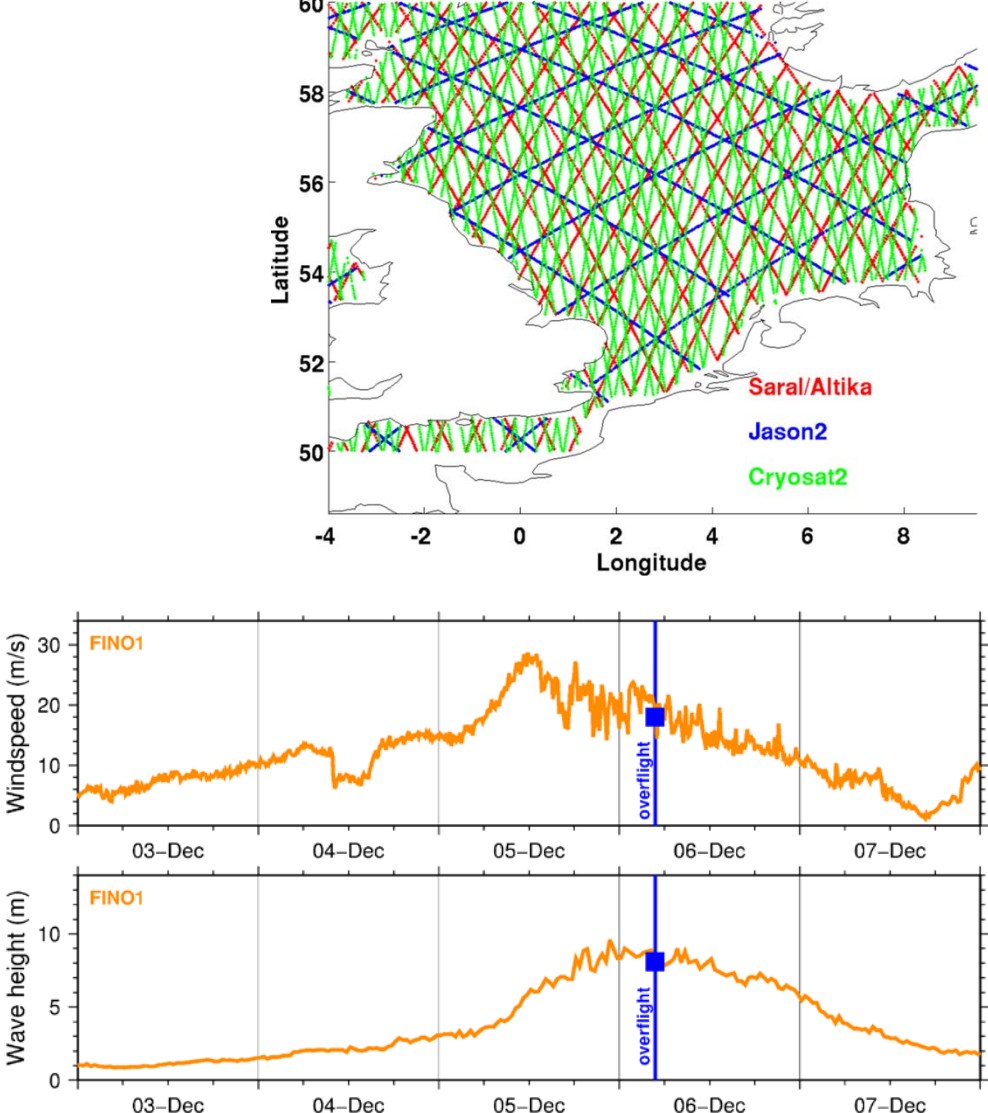

Figure 3: (top pattern) Tracks of all satellites during the study period; (middle and bottom pattern) Wind

speed *(middile) and wave height  (bottom) during  five days , which include the  Xaver storm at the*

*station FINO-1.*


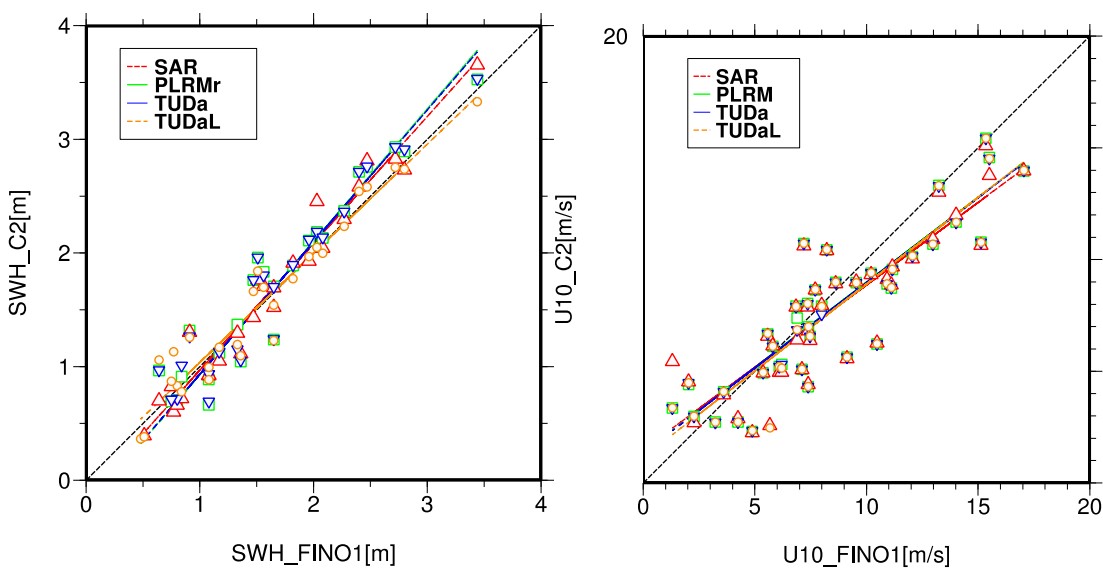


Figure 4. Comparison *of wave height (SWH, in m, left pattern) and wind speed (U10 in m7s, righ*

*patternt) of in-situ and CryoSat-2 altimeter data. at the station FINO-1.  Altimeter data used are DDA*

*altimetry  (SAR, triangle), standard PLRM (PLRMr and TUDA, square and inverse triangle) and*

*improved PLRM (TUDaL, circle).*

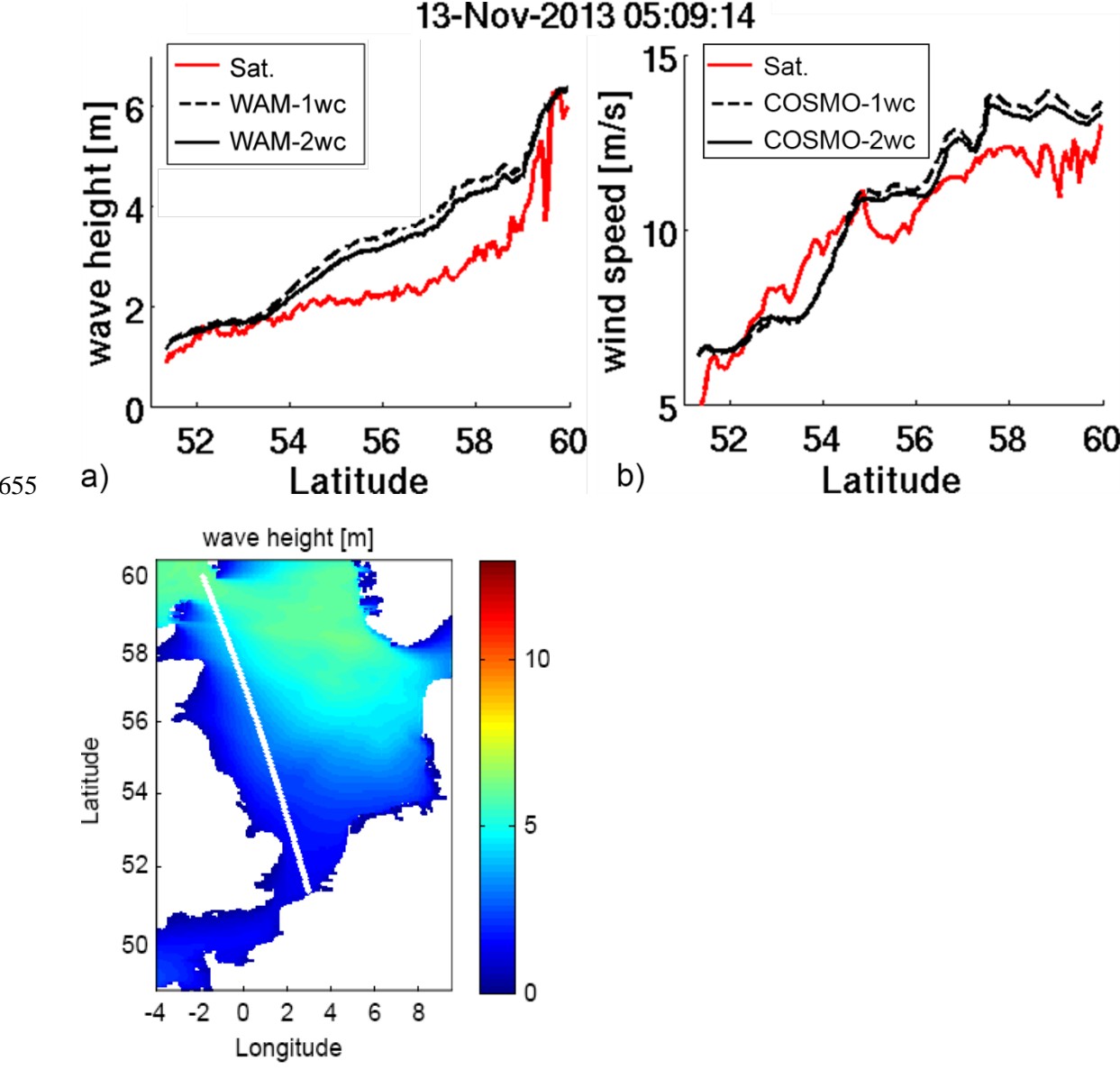

*Figure 5: Time series wave height (m) and wind speed (m/s) from the Saral/AltiKa data and as modelled by WAM-NS under calm weather conditions on 13 of November, 2013. The track of the satellite (the white line) is shown together with the model significant wave height at the time of the passage (bottom panel).*

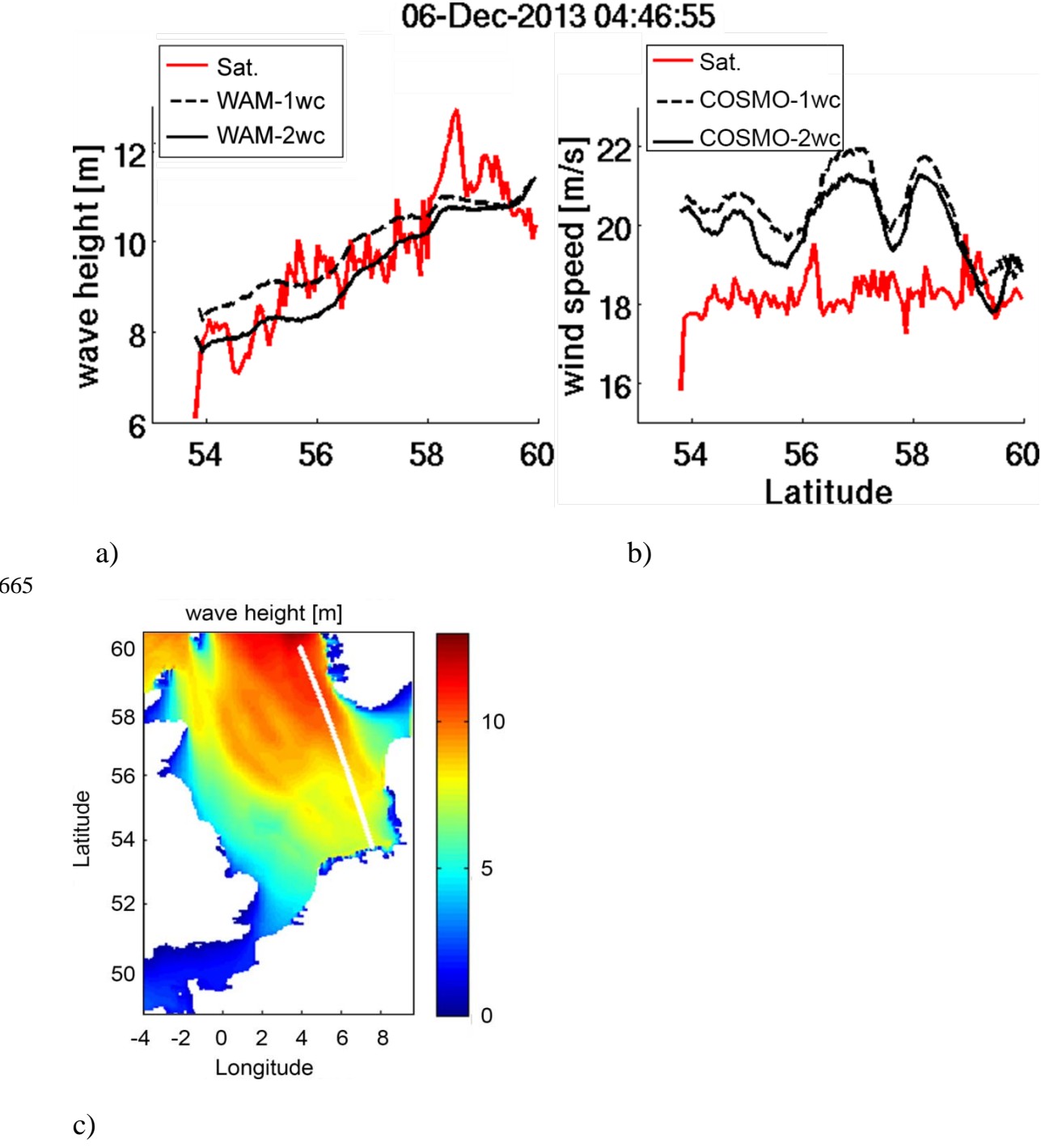


a)

b)

c)

*Figure 6 As Figure 5 but for the storm 'Xaver' on 06 December 2013..*

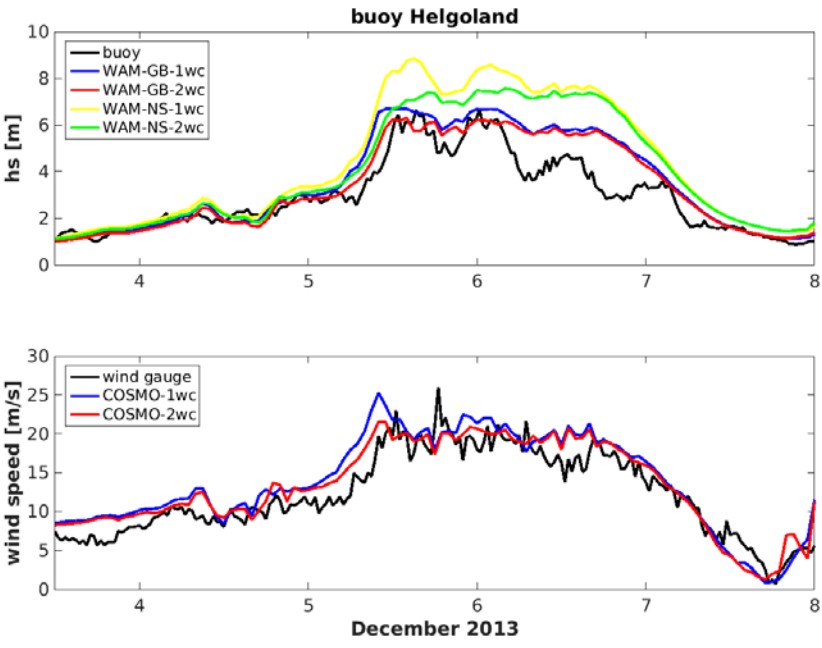

**(a)**

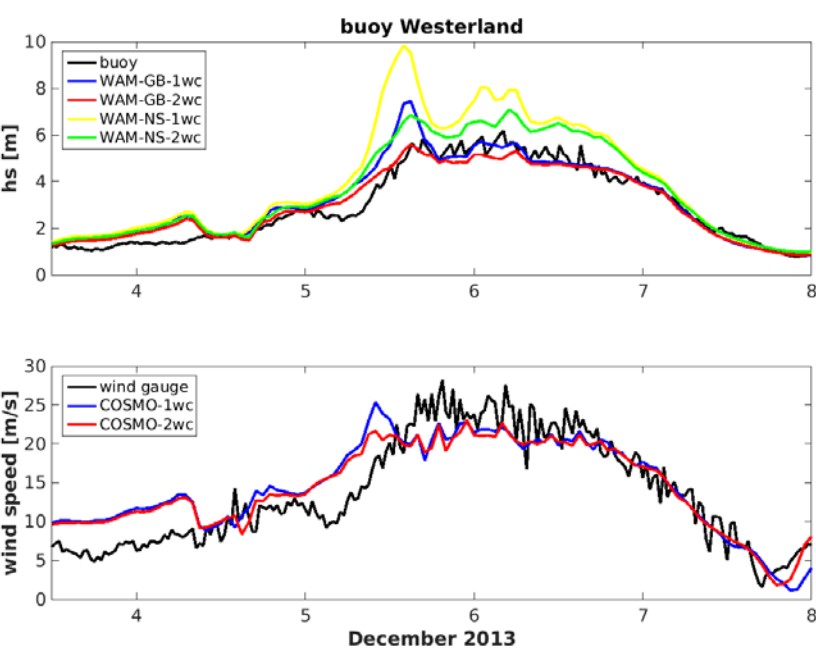

**(b)**

*Figure 7: (a,b) Significant wave height (m, top) and wind speed (m/s, bottom) during the storm 'Xaver' at the buoys Helgoland (a) and Westerland/Sylt (b).*

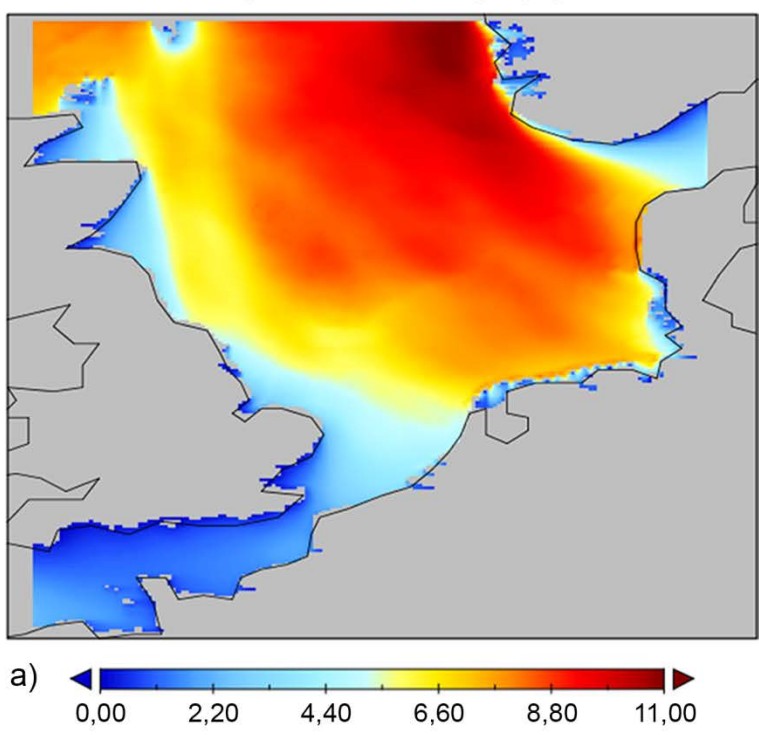

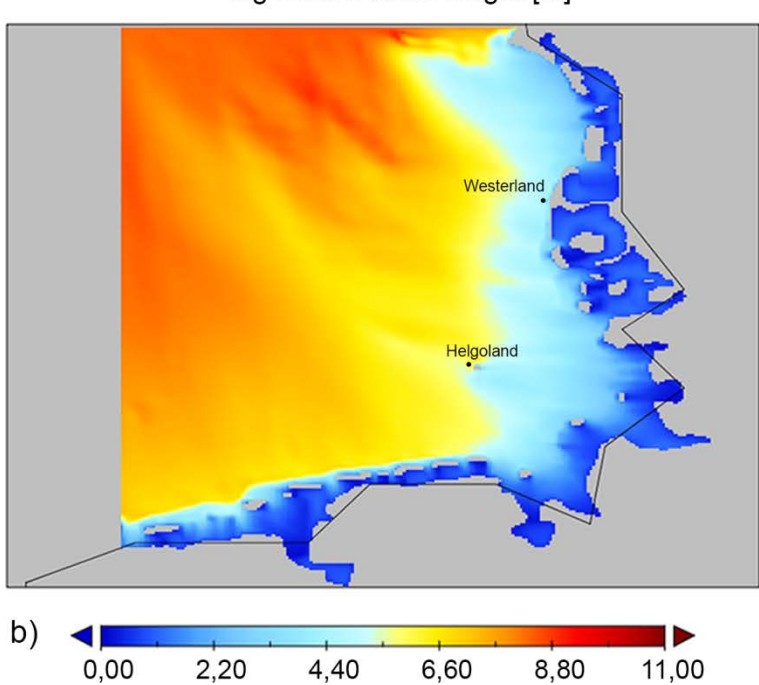


*Figure 8: (a,b) Significant wave height (m) in the North Sea (a) and the German Bight (b) at the peak of*

*the storm 'Xaver' (2013/12/6 9UTC) calculated by WAM-NS/GB-2wc.*

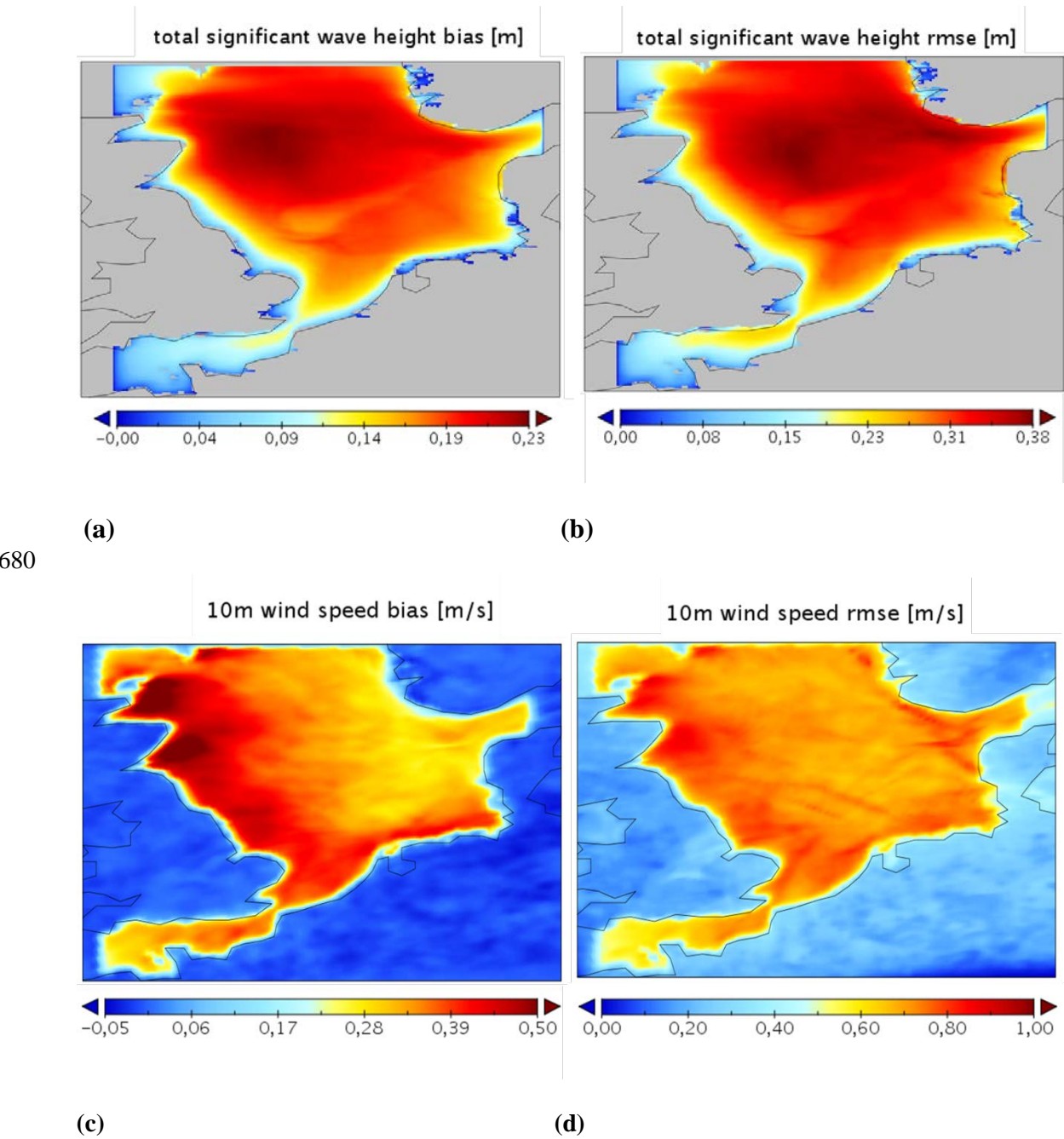


**(a)**                               **(b)**

**(c)**                               **(d)**

*Figure 9: (a,c)  Average difference (bias)  and (b,d)  rms difference (rmse)  of  WAM modeled significant wave height (m, top panel)  and COSMO modeled wind speed (m/s, bottom panel) when comparing one-way minus  two-way coupled modeling results. The differences are calculated as averaged over the whole three month period.*


mean sea level pressure [Pa]

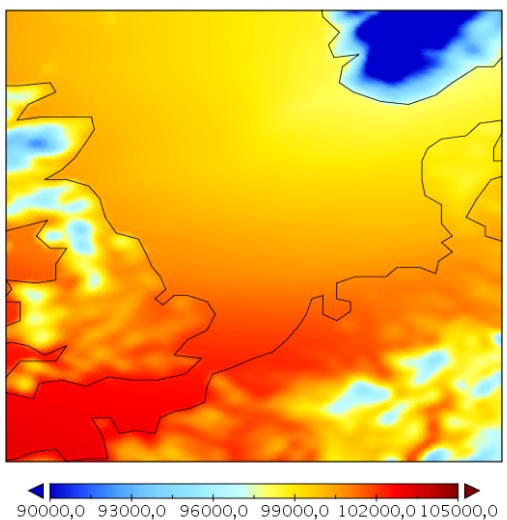

*(a)*

msl pressure difference [Pa]

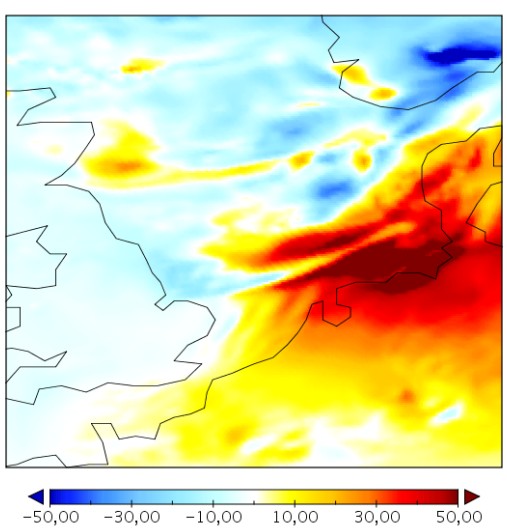

*(b)*

*Figure 10: (a) COSMO pressure (Pa) at mean sea level height in the North Sea during storm 'Xaver'*

*and (b) mean sea level pressure differences when comparing one-way minus two-way coupled modeling).*

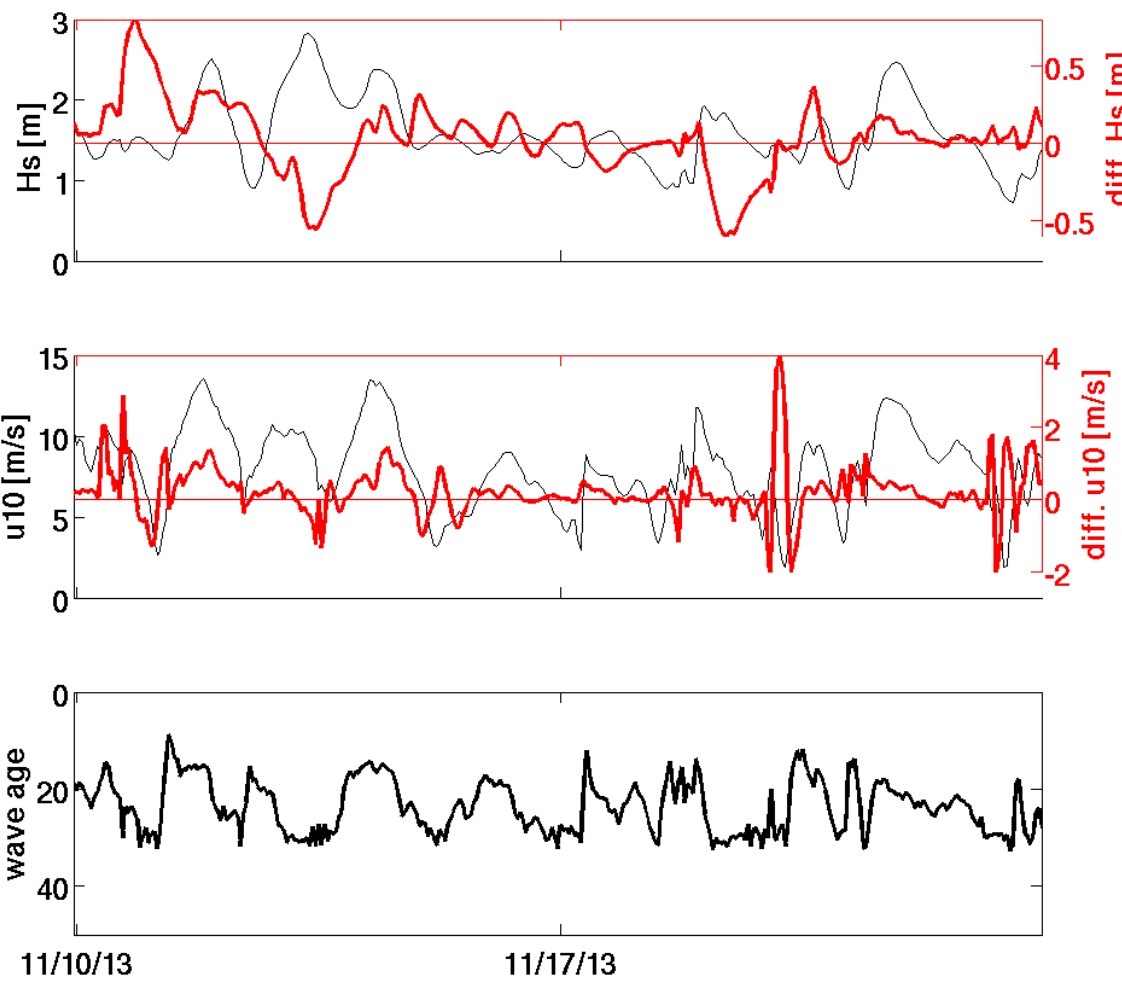

*Figure 11: Time series of significant wave height (m, top), wind speed (m/s, middle) and wave age (bottom) from the two-way coupled German Bight setup at FINO-1 for (a) a rather calm period with young wind sea and (b) during the storm 'Xaver'). Red lines in the top and middle panel show the differences between the one-way and the two-way coupled models.*


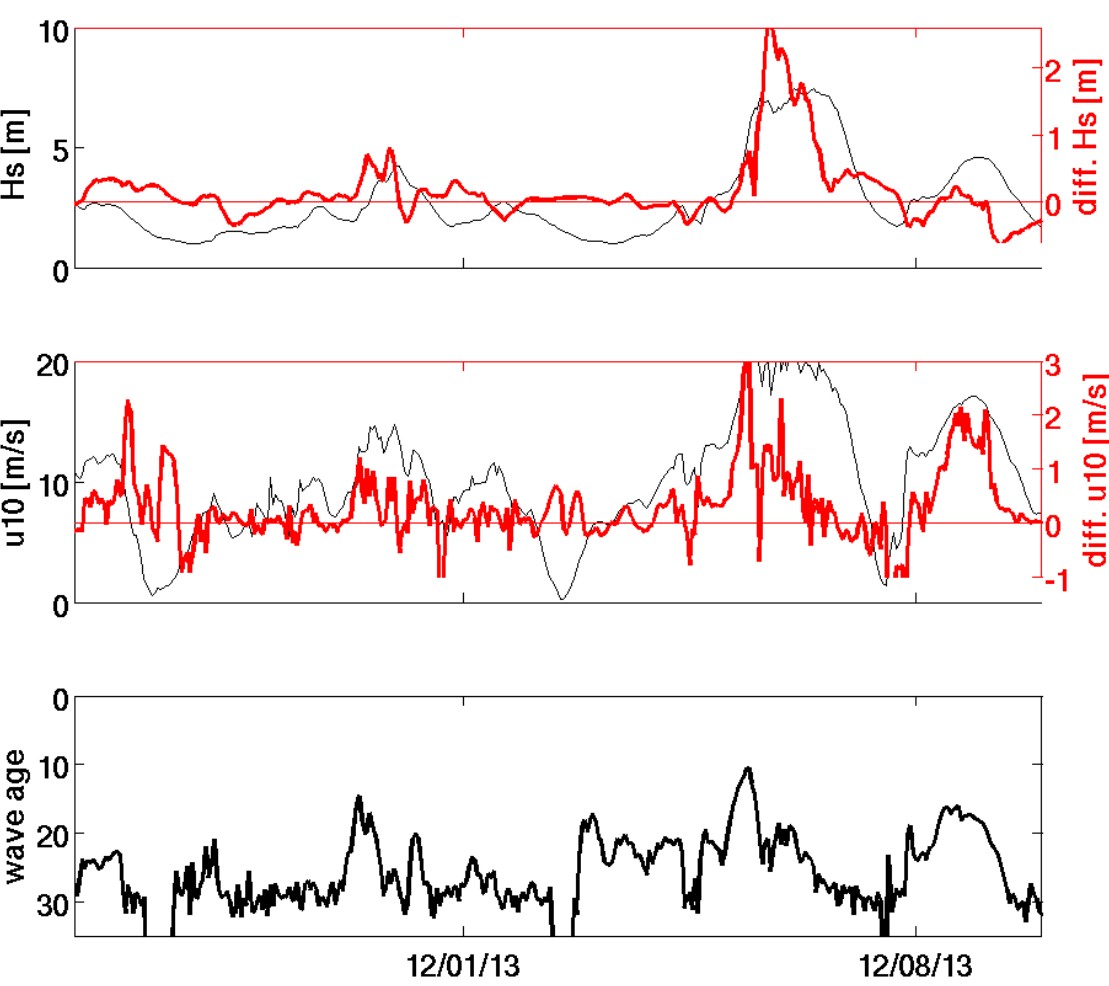

*Figure 12: As Figure 8 but during the storm 'Xaver'*
