# Peer review of "Wave-atmospheric modelling, satellite and *in situ* observations in the Southern North Sea: the impact of horizontal resolution and two-way coupling"

_Ocean Science, 2016_

## Referee Comment (RC1) · J.W. de Vries (Referee) · 29 Jul 2016

The subject of this manuscript is the 2-way coupling of atmospheric and wave models in the German Bight. Atmospheric forcing is supplied to the wave models, and the wave model sends back the surface roughness to the atmospheric model. The results of 2-way coupling are compared to 1-way coupling for a 3-month period that includes one of the most severe storms, named Xaver, in the last decades.

The subject is highly relevant and the technique promises a seizable improvement of wave forecasts in especially shallow areas with complex topography. And the authors indeed show an overall improvement of wave forecasts and also in particular for the Xaver storm.

It is, however, sometimes difficult to get to the message the authors try to convey. One of the reasons are the many errors in the english language, especially incorrect or missing articles, inconsistency between singular and plural, inconsistency in tenses, and missing commas. The manuscript should therefore be carefully checked and corrected.

Chapter 3, on results, is very fragmented and lacks a clear wrap-up and conclusion at the end. Moreover, especially Section 3.1 is very long and deals with a number of more or less separate items. It would help to puth these in separate subsections.

Sometimes the conclusions are contradictory to what the figures or tables suggest.

Comments in more detail:

1. Introduction

The reference Lionello (2003) is not in the references list. But the remark about what it states seems very odd here. The formulation suggests that the 2003 paper already describes the current work.

Towards the end, Staneva et al. (2016) is referenced. I would say that the subject of this paper has more relevance to the present paper than just the fact that wave heights are overestimated or the description of the used models. I would expect a discussion on coupling just waves and the atmosphere and including also circulation. And in the end you will probably want to couple all three together.

2.4 Integration Period and Data Availability

At the end, you refer to Figure 1b for the wave rider buoys, but they are in Figure 1c.

3.1 Validation of models

As in the final paper the tables will be close to the text, you might consider leaving out the values themselves here. It would make the text more easily readable.

Line 221: "due to the reasons explained above" is not clear which reasons you are aiming at.

Line 230: Change "It is well known that..." into "Passaro et al. (2014) established that...".

Line 240: "... wave heights are in good agreement." I do not agree. In the calm case, both models underestimate the wave height by approximately 1 m over a large part of the track.

Line 242: "... however,[!] the reduced wave height...". Change "however" into "although". Differences between the models are much smaller than the difference with the observations.

Line 246: The peak of the storm, at least the highest wave heights, are at the edge of the domain of the wave model. Any differences with observations will therefore strongly be influenced by the boundary conditions. And, actually, Figure 3b does not show a maximum, just an increasing wave height towards the North.

The comparison of these two tracks is a very useful illustration. You must have looked at the other tracks as well. Without giving any details here, it would be relevant to say something of the general picture that emerges from that. Does it agree with these two examples, or are there also other features there?

More or less the same remark on the comparison with the wave buoys. You show Helgoland and Westerland, but you should at least mention whether the results for Fino and Elbe are similar.

Line 280: As the results for both wave models are different in shallow water, what does that mean for the 2-way coupling? Should you not get the sea surface roughness from the model that includes wave breaking?

Line 289: "were provided by the DWD" is a remark that should be in Section 2.4.

Line 296: "Even though differences ... decrease ..." That is not what I see. Differences in biases in Table 2 are not really different between 50 and 100 m and the difference in standard deviation is even larger.

Line 303: The word "either" gives a choice between two possibilities. So it is not correct where you want to combine 4 things.

Line 307: It would help to give RMSE also in Table 3, if you refer to that instead of the standard deviation.

3.2 Impact

The first paragraph suggests that in 1-way coupling the coupling to the waves is too strong. If you would decrease this coupling, e.g. by a smaller Charnock constant, would that not give similar results for the waves? Then, what is the added value of the 2-way coupling?

You claim that the argument you infer from Figure 5 for wind speed differences is supported by the effect on wind stress (Figure 6). But the wind stress has a rather straightforward relation to the wind speed, so this is really the same argument. Just the fact that the wind stress is more that quadratic in the wind speed makes the effect only seem stronger.

Line 332: "... which tends to fill the low" is not what Janssen and Viterbo (1996) claim. They claim that the disturbance will grow less, what is not the same.

Line 336: "... indicates a shift of the pressure low minimum". That should be easily seen directly in the pressure fields. Why then an indirect argument?

Line 340: "such effects". What effects?

Figures:

The use of figures with several subfigures is not always an advantage. Some of the graphs, especially time series in Figures 3, 4 and 8 are already small and will be

smaller even and more difficult to read in a printed version. The authors might want to split some apart.

Figure 1: The explanation is not logical: first 1c and then 1b.

Figure 1b: I am not sure if a figure with all of the tracks is useful. They are not at the same time, and it is rather obvious that the pattern would look like that.

Figure 1c: The name Westerland is unreadable.

Figure 2: Units are missing on the wind speed scale.

Figure 3: The subfigures are not really similar enough to combine all of them. 3c and 3d could be taken together, but the way in which they are presented now suggests that 3a belongs to 3c and 3b to 3d.

As 3c and 3d are mentioned first in the text, they should be before 3a and 3b.

It would be useful to limit the area of Figure 3a to the same area as Figure 3b. Now a comparison is difficult.

It would help to indicate the buoys in 3a and 3b.

The color yellow in 3c and 3d is hardly visible.

Figure 5: The use of "bias" and "rmse" in this figure is confusing, as there terms are mostly used to indicate the difference with observations. Suggestion: "average difference" and "RMS difference". Also in Figure 6.

The figure might be explained more clearly. Either in the subscript, or in the text.

---

## Referee Comment (RC2) · Anonymous Referee #2 · 31 Jul 2016

Manuscript "An atmosphere-wave regional coupled model: improving predictions of wave heights and surface winds in the Southern North Sea by Kathrin Wahle et al. evaluates the effect of model coupling on the accuracy of modelled wave field in coastal areas. Model coupling especially for short-term forecasting purposes is a very topical issue and it is nice to see that the progress includes also coastal modelling. However, the authors state in several places that coupling of atmosphere and wave models is not novel in itself and that coupled models have been run operationally in many forecasting centres for decades. A reader would expect more detailed analysis of the effects of coupling on the coastal modelling, which is the novelty of this paper. Also, the analysis of the results should be done more carefully. In several places there are statements

that are not entirely supported by the Figures presented (cf. specific comments). The paper is fairly well structured, but the formulations and language require some further attention. I also recommend that the language is checked by a native speaker.

Some specific comments:

Section 1. Introduction: This section could be better structured and written. Explicit statements of what the authors are studying in this paper could be put in one place, preferably at the end of this section. Also the references to previous studies should be better formulated. Now it seems just a list of different coupled models presented in earlier studies. Please highlight their connection to the present study.

Section 2.3: Please give a short description of how the coupling was done, not just a reference to article by Ho-Hagemann et al.

Section 2.4, line 206: Here should probably be a reference to Fig. 1c, not 1b

Section 3.1, line 221: "reasons explained above" - Should it be "due to earlier explained reasons" and please give a reference to the section, where this explanation is given or explain it here.

Section 3.1: Did you compare the altimeter data against the Waveriders? How good is the accuracy of the altimeter data in the North Sea? And how was the match-up done between altimeter data and model data (distance in space and time, averaging, etc.)? What is the number of matched model-measured pairs for each altimeter and buoy?

Section 3.1, line 239-240: "In both cases measured and modelled wave heights are in good agreement" - is this really so? There seems to be quite big differences between the modelled and measured values along the track. Please be more precise.

Section 3.1, line 245-246: "the two-way coupled model results are closer to the mea-surements" - This is true for latitudes 54-55 and 57-58, but around latitude 56, the one-way coupled model seems to be closer to measurements. More detailed analysis is required.

[Figure]

**OSD**

Fig. 3: Why not use the same altimeter track to compare the performance on low-wind and storm conditions?

Section 3.1, lines 266-267: "Throughout the period WAM-NS-1wc shows the highest significant wave height" - This is true for Helgoland, but not for Westerland, where WAM-GB-1wc occasionally has higher values.

Section 3.1, lines 283-284 and Fig. 4d: What actually happens on December 5th in the Westerland in WAM-GB-1wc. Why is it behaving completely differently from the other setups? Nothing in the wind field seems to be supporting this kind of behaviour.

Figure 4: Would it be possible to mark the locations of the wave buoys to figures 4a and 4b. Although their locations are shown in Fig. 1, it would be easier for the reader to evaluate the model performance, if the locations would also be marked here.

Figure 8: Please use scales that show the whole range of the presented values of the chosen periods.

---

## Author Comment (AC1) · 8 Nov 2016

Answers of the Reviewer #1 comments

Dear Hans de Vries, Thank you for reviewing our manuscript and for the constructive comments and suggestions. In the revised manuscript your comments and suggestions for improvements have been carefully considered.

R#1: The subject of this manuscript is the 2-way coupling of atmospheric and wave models in the German Bight. Atmospheric forcing is supplied to the wave models, and the wave model sends back the surface roughness to the atmospheric model. The results of 2-way coupling are compared to 1-way coupling for a 3-month period

that includes one of the most severe storms, named Xaver, in the last decades. The subject is highly relevant and the technique promises a seizable improvement of wave forecasts in especially shallow areas with complex topography. And the authors indeed show an overall improvement of wave forecasts and also in particular for the Xaver storm.

Authors: Thank you for this nice appraisal.

R#1: It is, however, sometimes difficult to get to the message the authors try to convey. One of the reasons are the many errors in the english language, especially incorrect or missing articles, inconsistency between singular and plural, inconsistency in tenses, and missing commas. The manuscript should therefore be carefully checked and corrected.

Authors: The revised manuscript has been carefully checked by a native speaker, typos and errors in English language have been corrected.

R#1: Chapter 3, on results, is very fragmented and lacks a clear wrap-up and conclusion at the end. Moreover, especially Section 3.1 is very long and deals with a number of more or less separate items. It would help to put these in separate subsections. Sometimes the conclusions are contradictory to what the figures or tables suggest.

Authors: We agree and the revised manuscript has been re-structured. Chapter 3 with the results has been divided in two new Sections: Chapter 3 dealing with validation and Chapter 4 discussing the impact on wave-atmosphere coupling (one-way versus two-way). The new section 3 "Validation" has been also divided into three sub-sections describing separately and providing more analyses on the validation of altimeter data against in-situ data; model validations against satellite data and model validation against in-situ measurements. We think that the manuscript now is better structured.

R#1: Comments in more detail: 1. Introduction The reference Lionello (2003) is not

in the references list. But the remark about what it states seems very odd here. The formulation suggests that the 2003 paper already describes the current work.

Authors: The Introduction section has been re-structured and carefully revised (following also a similar comment of the Reviewer's #2). The state-of the art has been better presented. More information about the previous studies has been provided (incl. Lionello, 1998, 2003). The novelty in our work compared with previous studies has been described and argumentsfor performing our study has been presented.

R#1: Towards the end, Staneva et al. (2016) is referenced. I would say that the subject of this paper has more relevance to the present paper than just the fact that wave heights are overestimated or the description of the used models. I would expect a discussion on coupling just waves and the atmosphere and including also circulation. And in the end you will probably want to couple all three together.

Authors: e agree and in the revised manuscript a comprehensive discussion about the coupling between waves and hydrodynamics (referring also to our recent developments in Staneva et al. (2016) have been included. The perspectives and future plans towards implementing a fully coupled atmosphere-wave-circulation model, based on the developments and findings described here, as well as those by the wave-hydrodynamic coupling studies, have been discussed.

R#1: 2.4 Integration Period and Data Availability At the end, you refer to Figure 1b for the wave rider buoys, but they are in Figure 1c.

Authors: We apologize for this mistake and we refer to the right Figure in the revised manuscript. Even more, we removed from Figure 1, the figure showing the satellite tracks (old Fig.1b) and the new Figure 1 shows only the domain and bathymetry of the model areas.

R#1: 3.1 Validation of models As in the final paper the tables will be close to the text, you might consider leaving out the values themselves here. It would make the text

more easily readable.

Authors: We agree and left out the values that are given in the Table from the text while discussing the validations. We additionally reformulated this description in Section 3, making it clearer with additionally stressing on major conclusions from each sub-section.

R#1: Line 221: "due to the reasons explained above" is not clear which reasons you are aiming at.

Authors: We agree and changed accordingly. Similar comment has been also pointed at by the second reviewer, the revised text is re-phrased and we clearer refer to the introduction.

R#1: Line 230: Change "It is well known that..." into "Passaro et al. (2014) established that...".

Authors: The suggested revision has been made.

R#1: Line 240: "... wave heights are in good agreement." I do not agree. In the calm case, both models underestimate the wave height by approximately 1 m over a large part of the track.

Authors: The suggested revision has been made. We discussed the comparisons commented between the model and satellite wave heights and commented the discrepancies.

R#1: Line 242: "... however,[!] the reduced wave height...". Change "however" into "although". Differences between the models are much smaller than the difference with the observations.

Authors: The suggested revision has been made. We agree with this comment – the differences between the model runs are most significant and indeed smaller than between model and observations. This has been now revised in Section 3 and also

stressed in the discussion section.

R#1: Line 246: The peak of the storm, at least the highest wave heights, are at the edge of the domain of the wave model. Any differences with observations will therefore strongly be influenced by the boundary conditions. And, actually, Figure 3b does not show a maximum, just an increasing wave height towards the North.

Authors: We agree with the comment and the discussion of the comparisons (Fig.6a in the revised manuscript) has been re-phrased, making it clearer. The maximum that we referred is observed by the satellite data. Indeed for the both model runs the wave height increases northward. This has been additionally commented in Section 3.2.

R#1: The comparison of these two tracks is a very useful illustration. You must have looked at the other tracks as well. Without giving any details here, it would be relevant to say something of the general picture that emerges from that. Does it agree with these two examples, or are there also other features there?

Authors: We looked also at many other tracks and the results were similar to the ones for the calm situation. In general, the measured wind speeds were in slightly better agreement with the modeled ones as can be seen also from the statistics in Table 1. The track for storm 'Xaver' was the only one taken under such extreme conditions. A discussion on this has been added in Section 3.2.

R#1: More or less the same remark on the comparison with the wave buoys. You show Helgoland and Westerland, but you should at least mention whether the results for Fino and Elbe are similar.

Authors: We have also looked at the comparisons between in-situ and modelled data for Fino and Westerland stations. Those comparisons look similar to what we show. The general picture and the conclusions agree. We added additional discussion on that, as well in Section 3.3.

R#1: Line 280: As the results for both wave models are different in shallow water, what

does that mean for the 2-way coupling? Should you not get sea surface roughness from the model that includes wave breaking?

Authors: We agree with the comment that it is difficult to differentiate between effects coming from wave breaking and from two-way coupling. This has been thoughtfully discussed in Section 3.3 and additional references have been provided.

R#1: Line 289: "were provided by the DWD" is a remark that should be in Section 2.4.

Authors: The suggested revision has been made.

R#1: Line 296: "Even though differences ... decrease ..." That is not what I see. Differences in biases in Table 2 are not really different between 50 and 100 m and the difference in standard deviation is even larger.

Authors: We apologize for this incorrectness and fully agree with the comment. Only the bias slightly decreases. The description of those results has been corrected in the revised manuscript and we made the explanations clearer.

R#1: Line 303: The word "either" gives a choice between two possibilities. So it is not correct where you want to combine 4 things.

Authors: The suggested revision has been made.

R#1: Line 307: It would help to give RMSE also in Table 3, if you refer to that instead of the standard deviation.

Authors: That was a mistake in writing and we changed 'rmse' into 'standard deviation' in the text.

R#1: 3.2 Impact The first paragraph suggests that in 1-way coupling the coupling to the waves is too strong. If you would decrease this coupling, e.g. by a smaller Charnock constant, would that not give similar results for the waves? Then, what is the added value of the 2-way coupling?

Authors: We agree that by calibrating the parameters one can achieve better results compared to the measurements even using only a one-way coupled experiment (e.g. as in Zweers et al., 2002). This has been discussed in the instruction in the revised manuscript. However, the aim of our work was also to perform a process oriented study by considering the feedback of the bottom roughness by the sea state dependence of the surface stress via the two-way atmosphere-wave coupling. These developments were needed for future extension of the coupled model system by integrating atmosphere-wave–current interactions to further investigate the effects of coupling, especially during extreme storm events.

R#1: You claim that the argument you infer from Figure 5 for wind speed differences is supported by the effect on wind stress (Figure 6). But the wind stress has a rather straightforward relation to the wind speed, so this is really the same argument. Just the fact that the wind stress is more that quadratic in the wind speed makes the effect only seem stronger.

Authors: We agree and the suggested revision has been made. In the revised manuscript we removed the redundant figure with the horizontal patterns of wind stress rmse and bias (Figure 6 in the first submission).

R#1: Line 332: "... which tends to fill the low" is not what Janssen and Viterbo (1996) claim. They claim that the disturbance will grow less, what is not the same.

Authors: We agree and this has been re-phrased in the revised manuscript.

R#1: Line 336: "... indicates a shift of the pressure low minimum". That should be easily seen directly in the pressure fields. Why then an indirect argument?

Authors: We agree with the comment. More information has been added in the discussion of the pressure filed in Section 4.

R#1: Line 340: "such effects". What effects?

Authors: The suggested revision has been made. We changed this into 'to observe

this impact'.

R#1: Figures: The use of figures with several subfigures is not always an advantage. Some of the graphs, especially time series in Figures 3, 4 and 8 are already small and will be smaller even and more difficult to read in a printed version. The authors might want to split some apart.

Authors: We agree and the organization of the figures has been changed in the revised manuscript. The sub-plots have been split in different figures. The figure patterns were made larger. Additionally, the quality of the individual figures has been improved. Some of them have been re-ordered following the logics in the text.

R#1: Figure 1: The explanation is not logical: first 1c and then 1b. Figure 1b: I am not sure if a figure with all of the tracks is useful. They are not at the same time, and it is rather obvious that the pattern would look like that. Figure 1c: The name Westerland is unreadable.

Authors: We agree – actually we removed Figure 1b with the satellite tracks from Figure 1. The size of the figures has been changed and their quality improved.

R#1: Figure 2: Units are missing on the wind speed scale.

Authors: The suggested revision has been made. We added the missing units to the figures caption and in the figure.

R#1: Figure 3: The subfigures are not really similar enough to combine all of them. 3c and 3d could be taken together, but the way in which they are presented now suggests that 3a belongs to 3c and 3b to 3d. As 3c and 3d are mentioned first in the text, they should be before 3a and 3b. It would be useful to limit the area of Figure 3a to the same area as Figure 3b. Now a comparison is difficult. It would help to indicate the buoys in 3a and 3b. The color yellow in 3c and 3d is hardly visible.

Authors: The suggested revision has been made. We made two separate figures out of it: first one with the comparison of measurements and model results and a second

one with the distribution of Hs in the model area. We also separated the figures for the different events. The quality and presentation of the figures (lines, colors, fonts) has been improved. The Figures have been then re-numbered.

R#1: Figure 5: The use of "bias" and "rmse" in this figure is confusing, as there terms are mostly used to indicate the difference with observations. Suggestion: "average difference" and "RMS difference". Also in Figure 6. The figure might be explained more clearly. Either in the subscript, or in the text.

Authors: The suggested revision has been made. We introduced the terms as suggested in the text and in the figures caption and tried to explain more clearly what is shown in the figures caption.

―――――――――――――――――――

---

## Author Response (AR1)

**Authors Response to the Reviewers' comments**

**Rev.# 1**

**J.W. de Vries (Referee)**
hans.de.vries@knmi.nl

Dear Hans de Vries,
Thank you for reviewing our manuscript and for the constructive comments and suggestions. In the revised manuscript your comments and suggestions for improvements have been carefully considered.

The subject of this manuscript is the 2-way coupling of atmospheric and wave models in the German Bight. Atmospheric forcing is supplied to the wave models, and the wave model sends back the surface roughness to the atmospheric model. The results of 2-way coupling are compared to 1-way coupling for a 3-month period that includes one of the most severe storms, named Xaver, in the last decades.
The subject is highly relevant and the technique promises a seizable improvement of wave forecasts in especially shallow areas with complex topography. And the authors indeed show an overall improvement of wave forecasts and also in particular for the Xaver storm.

Authors: Thank you for this nice appraisal.

It is, however, sometimes difficult to get to the message the authors try to convey. One of the reasons are the many errors in the english language, especially incorrect or missing articles, inconsistency between singular and plural, inconsistency in tenses, and missing commas. The manuscript should therefore be carefully checked and corrected.

Authors:
The revised manuscript has been carefully checked by a native speaker; typos and errors in the English language have been corrected.

Chapter 3, on results, is very fragmented and lacks a clear wrap-up and conclusion at the end. Moreover, especially Section 3.1 is very long and deals with a number of more or less separate items. It would help to put these in separate subsections.
Sometimes the conclusions are contradictory to what the figures or tables suggest.

Authors:
We agree and the revised manuscript has been re-structured. Chapter 3 with the results has been spitted in two new Sections: Chapter 3 dealing with validation and Chapter 4 in which we discuss the impact on wave-atmosphere coupling (one-way versus two-way). The new Section 3: "Validation" has also been divided into three sub-sections describing separately and providing more analyses on 3.1: the validation of altimeter data against in situ data (this is a completely new part in the revised manuscript); 3.2: model validations against satellite data and 3.3: model

validation against in situ measurements. We think that the manuscript is now better structured and reads easier.

50 Comments in more detail:

1. Introduction
The reference Lionello (2003) is not in the references list. But the remark about what it states seems very odd here. The formulation suggests that the 2003 paper already
55 describes the current work.

Authors: The Introduction section has been re-structured and carefully revised (following also a similar comment of the Reviewer's #2). The state-of the art has been better presented. More information about the previous studies has been provided (incl. Lionello, 1998, 2003). The novelty
60 in our work compared with previous studies has been described and arguments for performing our study have been presented.

Towards the end, Staneva et al. (2016) is referenced. I would say that the subject of this paper has more relevance to the present paper than just the fact that wave heights
65 are overestimated or the description of the used models. I would expect a discussion on coupling just waves and the atmosphere and including also circulation. And in the end you will probably want to couple all three together.

Authors: We agree and in the revised manuscript a comprehensive discussion about the coupling
70 between waves and hydrodynamics (referring also to our recent developments in Staneva et al., 2016) have been included. The perspectives and future plans towards implementing a filly coupled atmosphere-wave-circulation model, based on the developments and findings described here, as well as those by the wave-hydrodynamic coupling studies, have been discussed.

75 2.4 Integration Period and Data Availability
At the end, you refer to Figure 1b for the wave rider buoys, but they are in Figure 1c.

Authors: We apologize for this mistake, and we refer to the correct figure in the revised manuscript. Additionally, we removed the satellite tracks from Figure 1 (old Fig. 1b). The new
80 Figure 1 shows only the domain and bathymetry of the model areas.

3.1 Validation of models
As in the final paper the tables will be close to the text, you might consider leaving out the values themselves here. It would make the text more easily readable.
85
Authors: We agree and removed the values that are given in the Table from the text in the discussion of the validations. We additionally reformulated this description in Section 3, making it clearer by stressing on major conclusions from each sub-section.

90 Line 221: "due to the reasons explained above" is not clear which reasons you are aiming at.

Authors: We agree and changed the text accordingly. Similar comment has also been given mentioned by the second reviewer, the revised text was re-phrased and we clearly refer to the
95 introduction part.

Line 230: Change "It is well known that..." into "Passaro et al. (2014) established that...".

Authors: The suggested revision has been made.

Line 240: "... wave heights are in good agreement." I do not agree. In the calm case, both models underestimate the wave height by approximately 1 m over a large part of the track.

Authors: The suggested revision has been made. We discussed the comparisons between the model and satellite altimetry wave heights and commented on the discrepancies in Section 3.2.

Line 242: "... however,[!] the reduced wave height...". Change "however" into "although". Differences between the models are much smaller than the difference with the observations.

Authors: The suggested revision has been made. We agree with this comment – the differences between the model runs are most significant and indeed smaller than between model and observations. This has been now revised in Section 3 and also stressed in the discussion section.

Line 246: The peak of the storm, at least the highest wave heights, are at the edge of the domain of the wave model. Any differences with observations will therefore strongly be influenced by the boundary conditions. And, actually, Figure 3b does not show a maximum, just an increasing wave height towards the North.

Authors: We agree with the comment and the discussion of the comparisons ( the results in Fig. 6 a in the revised manuscript) has been re-phrased, making it clearer. The maximum that we referred is observed by the satellite data. Indeed for the both model runs, the wave height increases northward. This finding has been commented on in Section 3.2.

The comparison of these two tracks is a very useful illustration. You must have looked at the other tracks as well. Without giving any details here, it would be relevant to say something of the general picture that emerges from that. Does it agree with these two examples, or are there also other features there?

Authors: We looked at many other tracks, and the results were similar to the ones for the calm situation. In general, the measured wind speeds were in slightly better agreement with the modelled ones as seen also from the statistics in Table 1. The track for storm Xaver was the only one taken under such extreme conditions. This discussion has been added in Section 3.2.

More or less the same remark on the comparison with the wave buoys. You show Helgoland and Westerland, but you should at least mention whether the results for Fino and Elbe are similar.

Authors: We have also looked at the comparisons between in situ and modelled data for FINO-1 and Elbe stations. Those comparisons look similar to what we show. The general picture and the conclusions agree. We added additional discussion to Section 3.3.

Line 280: As the results for both wave models are different in shallow water, what does

that mean for the 2-way coupling? Should you not get sea surface roughness from
the model that includes wave breaking?

150 Authors: We agree with the comment that it is difficult to differentiate between effects coming from wave breaking and from two-way coupling. This point has been thoughtfully discussed in Section 3.3 and additional references have been provided.

Line 289: "were provided by the DWD" is a remark that should be in Section 2.4.

155 Authors: The suggested revision has been made.

Line 296: "Even though differences ... decrease ..." That is not what I see. Differences
in biases in Table 2 are not really different between 50 and 100 m and the difference in
standard deviation is even larger.

160 Authors: We apologize for this incorrectness and fully agree with the comment. Only the bias slightly decreases. The description of those results has been corrected in the revised manuscript, and we made the explanations clearer.

Line 303: The word "either" gives a choice between two possibilities. So it is not correct
165 where you want to combine 4 things.

Authors: The suggested revision has been made.

Line 307: It would help to give RMSE also in Table 3, if you refer to that instead of the
170 standard deviation.

Authors: That was a mistake in writing and we changed 'rmse' into 'standard deviation' in the text.

3.2 Impact
175 The first paragraph suggests that in 1-way coupling the coupling to the waves is too
strong. If you would decrease this coupling, e.g. by a smaller Charnock constant,
would that not give similar results for the waves? Then, what is the added value of the
2-way coupling?

180 Authors:
We agree that by calibrating the parameters one can achieve better results compared to the
measurements even using only a one-way coupled experiment (e.g., as in Zweers et al., 2002). This
has been discussed in the introduction in the revised manuscript. However, the aim of our work
was also to perform a process oriented study by considering the feedback of the surface roughness
185 by the sea state dependence of the surface stress via the two-way atmosphere-wave coupling.
These developments  are needed for future extension of the coupled model system by integrating
atmosphere-wave–current interactions to further investigate the effects of coupling, especially
during extreme storm events.

190 You claim that the argument you infer from Figure 5 for wind speed differences is supported
by the effect on wind stress (Figure 6). But the wind stress has a rather straightforward

relation to the wind speed, so this is really the same argument. Just the fact
that the wind stress is more that quadratic in the wind speed makes the effect only
seem stronger.

195

Authors: We agree and the suggested revision has been made. In the revised manuscript we
removed the redundant figure with the horizontal patterns of wind stress rmse and bias (Figure 6
in the first submission).

200     Line 332: "... which tends to fill the low" is not what Janssen and Viterbo (1996) claim.
They claim that the disturbance will grow less, what is not the same.

Authors: We agree and this has been re-phrased in the revised manuscript.

205     Line 336: "... indicates a shift of the pressure low minimum". That should be easily
seen directly in the pressure fields. Why then an indirect argument?

Authors: We agree with the comment. More information has been added in the discussion of the
pressure filed in Section 4.

210

Line 340: "such effects". What effects?

Authors: The suggested revision has been made. We changed this into 'to observe this impact'.

215     Figures:
The use of figures with several subfigures is not always an advantage. Some of the
graphs, especially time series in Figures 3, 4 and 8 are already small and will be smaller even and
more difficult to read in a printed version. The authors might want to
split some apart.

220

Authors: We agree and the organization of the figures has been changed in the revised manuscript.
The sub-plots have been split in different figures. The figure patterns were made larger.
Additionally, the quality of the individual figures has been improved. Some of them have been re-
ordered following the logics in the text.

225

Figure 1: The explanation is not logical: first 1c and then 1b.
Figure 1b: I am not sure if a figure with all of the tracks is useful. They are not at the
same time, and it is rather obvious that the pattern would look like that.
230     Figure 1c: The name Westerland is unreadable.

Authors: We agree – actually we removed Figure 1b with the satellite tracks from Figure 1. The size
of the figures has been changed and their quality improved.

235     Figure 2: Units are missing on the wind speed scale.

Authors: The suggested revision has been made. We added the missing units to the figures caption
and in the figure.

240     Figure 3: The subfigures are not really similar enough to combine all of them. 3c and
3d could be taken together, but the way in which they are presented now suggests that

3a belongs to 3c and 3b to 3d.
As 3c and 3d are mentioned first in the text, they should be before 3a and 3b.
It would be useful to limit the area of Figure 3a to the same area as Figure 3b. Now a
comparison is difficult.
It would help to indicate the buoys in 3a and 3b.
The color yellow in 3c and 3d is hardly visible.

Authors: The suggested revision has been made. We made two separate figures out of it: first one
with the comparison of measurements and model results and a second one with the distribution of
Hs in the model area. We also separated the figures for the different events.
The quality and presentation of the figures (lines, colors, fonts) has been improved. The Figures
have been then re-numbered.

Figure 5: The use of "bias" and "rmse" in this figure is confusing, as there terms are
mostly used to indicate the difference with observations. Suggestion: "average difference"
and "RMS difference". Also in Figure 6.
The figure might be explained more clearly. Either in the subscript, or in the text.

Authors: The suggested revision has been made. We introduced the terms as suggested in the text
and in the figures caption and tried to explain more clearly what is shown in the figures caption.

**Rev.# 2**

Authors: Thank you for the review of our manuscript.
265    We appreciate the constructive comments and will revise the manuscript in accordance with the reviewer's comments.

Manuscript "An atmosphere-wave regional coupled model: improving predictions of wave heights and surface winds in the Southern North Sea by Kathrin Wahle et al.
270    evaluates the effect of model coupling on the accuracy of modelled wave field in coastal areas. Model coupling especially for short-term forecasting purposes is a very topical issue and it is nice to see that the progress includes also coastal modelling. However, the authors state in several places that coupling of atmosphere and wave models is not novel in itself and that coupled models have been run operationally in many forecasting
275    centres for decades. A reader would expect more detailed analysis of the effects of coupling on the coastal modelling, which is the novelty of this paper. Also, the analysis of the results should be done more carefully. In several places there are statements that are not entirely supported by the Figures presented (cf. specific comments). The paper is fairly well structured, but the formulations and language require some further
280    attention. I also recommend that the language is checked by a native speaker.

Authors: We agree. The manuscript has been revised and the analyses of the model results were more precisely presented. Deeper discussion and more information are provided of the role of two-way coupling on the coastal model results. Following the similar comments of the Reviewer #1 this
285    Section has been re-organized (see also the answers to Reviewer #1 comments about that).

The revised manuscript has been carefully checked by a native speaker, typos and errors in English language have been corrected.

290    Some specific comments:
Section 1. Introduction: This section could be better structured and written. Explicit statements of what the authors are studying in this paper could be put in one place, preferably at the end of this section. Also the references to previous studies should be better formulated. Now it seems just a list of different coupled models presented in
295    earlier studies. Please highlight their connection to the present study.

Authors: We agree with the comments and the suggested revisions have been done. The introduction part has been re-organized following this suggestion. Earlier works have been better formulated and new references added. The discussion on what we are studying in this manuscript,
300    stressing on the novelty compared to the early studies is given now in one paragraph.

Section 2.3: Please give a short description of how the coupling was done, not just a reference to article by Ho-Hagemann et al.

305    Authors: The suggested revision has been done in Section 2.3 of the revised manuscript.

Section 2.4, line 206: Here should probably be a reference to Fig. 1c, not 1b

Authors: We apologize for the mistake and refer now to the correct figure.  The old Fig.1b has been
310    removed from Figure 1.

Section 3.1, line 221: "reasons explained above" - Should it be "due to earlier explained reasons" and please give a reference to the section, where this explanation is given or explain it here.

315

Authors: We agree. The text has been modified and we provided clearer statements.

Section 3.1: Did you compare the altimeter data against the Waveriders? How good is the accuracy of the altimeter data in the North Sea? And how was the match-up done
320 between altimeter data and model data (distance in space and time, averaging, etc.)?

Authors: We completely agree that additional information is needed. In the revised manuscript we introduce reference to previous studies in the introduction. We also included a new sub-section (now Section 3.1) dealing with comparisons of altimeter data against in-situ measurements to
325 estimate the accuracy of altimeter data. Analyses and new figures demonstrating these comparisons have been also included in the revised version.
We also included a discussion about the match-up of altimeter and model data in our study in Section 3.2.

330 What is the number of matched model-measured pairs for each altimeter and buoy?

Authors: For the altimeter data these numbers are given in Table1 (about 7000 for each of them). For the buoys there are about 4000 matched pairs. These numbers have been also now given in Sections 3.2 and 3.3, correspondingly, of the revised manuscript.

335
Section 3.1, line 239-240: "In both cases measured and modelled wave heights are in good agreement" - is this really so? There seems to be quite big differences between the modelled and measured values along the track. Please be more precise.

340 Authors: The suggested revisions have been done. The validation results have been more precisely discussed in Section 3.2 and additional arguments and explanations presented.

Section 3.1, line 245-246: "the two-way coupled model results are closer to the measurements" - This is true for latitudes 54-55 and 57-58, but around latitude 56, the
345 one-way coupled model seems to be closer to measurements. More detailed analysis is required.

Authors: We agree with the comment about the discussion of the comparisons with the measurements. More detailed analyses have been added in Section 3.2 about the spatial
350 (latitudinal) distribution of the satellite date and model simulations. Critical discussion on the results has been included.

Fig. 3: Why not use the same altimeter track to compare the performance on low-wind and storm conditions?
355

Authors: The choice of tracks to compare the performance (Figure 5 and Figure 6 in the revised manuscript), also following the comments of the reviewer #1, has been discussed in Section 3.2 in the revised manuscript.

360 Section 3.1, lines 266-267: "Throughout the period WAM-NS-1wc shows the highest significant wave height" - This is true for Helgoland, but not for Westerland, where

WAM-GB-1wc occasionally has higher values.

Authors: We agree with the comment and this has been added in the revised manuscript. Additional
analyses are provided making now the description in Section 3 more precisely.

Section 3.1, lines 283-284 and Fig. 4d: What actually happens on December 5th in the
Westerland in WAM-GB-1wc. Why is it behaving completely differently from the other
setups? Nothing in the wind field seems to be supporting this kind of behaviour.

Authors: We apologize for the mistake, which we made while plotting WAM-GB-1wc run (blue line
in Fig.4). The figure has been plotted correctly (Figure 7 in the revised manuscript).

Figure 4: Would it be possible to mark the locations of the wave buoys to figures 4a
and 4b. Although their locations are shown in Fig. 1, it would be easier for the reader
to evaluate the model performance, if the locations would also be marked here.

Authors: Following the Reviewer #1 suggestions, the organization of the figures has been changed
in the revised manuscript. The different sub-plots have been split into separate figures. The
patterns in the new Figures were made larger. Additionally, the quality of the individual figures has
been improved. Some of them have been re-ordered following the logics in the text. Figure 4 a,b is
Figure 8 and Fig. 4c,d – Fig. 7.

Figure 8: Please use scales that show the whole range of the presented values of the
chosen periods.

Authors: As described in the text the wave age is calculated as the quotient of phase and friction
velocity it gets very high values by calm weather conditions when the wind speed is very low.
Therefore, we introduced a wave age limit in the plots making the time variability and the ranges
clearer. Additional information and description on the wave age plots has been added in Section 4.

[revised manuscript text omitted]

---

## Referee Report (RR1)

Revised manuscript "Wave-atmospheric modelling, satellite and in situ observations in the Southern North Sea: the impact of horizontal resolution and two-way coupling" by Kathrin Wahle et al. presents interesting results of how two-way atmosphere-wave coupling improves the  coastal wave forecast. This is an important step in development of coupled models for short-term coastal forecasting with high-resolution, especially as Authors have been able to show that the very high-resolution coastal applications also benefit from the two-way coupling. The revision has improved the manuscript, making it more focused and easier to read. The Authors have also well taken into account the reviewers comments and suggestions. There are, however, still few places where further clarification is needed (please see my detailed comments below).  Also, I suggest that some copy editing is done to the text before publication. Some sentences are quite long and therefore not easy to follow and there is also some repetition.

Some detailed comments:

Section 1. Introduction: This section has improved a lot, and is now better structured and easier to follow. It is, however, quite long and has repetition. I think it would benefit of some copy-editing.

Section 1, lines 58-59: alternative to fully-coupled ocean-atmosphere model? Would this model include also waves? And why should we have alternatives, shouldn't we aim for the fully coupled models.

Section 2.2, line 169: Should it be WAM4.5.4?

Section 3.1, line 260: the situ-data → the in-situ data?

Section 3.2, first paragraph: The bias is calculated as measured minus modelled value and following this, in text it is said that altimeter data underestimates the modelled values. Shouldn't this be said the other way around? E.g. modelled values are overestimated compared to the satellite measurements. In  most of the cases I'd assume the altimeter data to be more accurate than the modelled data and it is said to be the dataset against which the model is verified.

Section 3.2, first paragraph: If the Authors do not trust the Cryosat-swh, why is it used for validation in the first place. Wouldn't two altimeter datasets be enough for validation? Anyhow, the explanation related to this in lines 287-297 is bit complicated to follow. Should the reader disregard the results from this comparison or interpret them with care?

Section 3.2, lines 285-287: Quite a long sentence, could be split to to parts

Section 3.3, second paragraph: The reason behind the better behaviour of the high-resolution model is probably mostly due to the better description of the bathymetry in the area, not the high-resolution *per se.* It is implicitly mentioned by the Authors, but it could be stated more clear.

Section 4, line 414: Why not simply say, that the fetch is too short for the waves to evolve.

Section 5, lines 446-449: Quite a long sentence, could be split to two parts.

Section 5, lines 473-474: What is meant by potential uncertainties of shallow water in the wave model? Is this related to the description of bathymetry or to the wave model source terms related to shallow water physics?

Tables 1-3: Table captions should explain what the red and green colouring means.

Figure 3: Please explain the marked overflight also in the Figure caption.

In the Figure and table captions there is a mixture of terms "wave height" and "significant wave height". Preferably "significant wave height" should be used in all of them.

---

## Referee Report (RR2)

The manuscript has certainly improved, but there is still work left.

**Title**

I liked the old title better than the new one. The title should convey the essence of the paper, and for me that is the effect of a two-way coupling of an atmosphere and wave model on the representation of atmospheric and wave parameters in a shallow and complex coastal area. Explicitly mentioning both in-situ and satellite observations draws to much attention away from that. And, moreover, the title is now too much a collection of loose terms without really connecting them.

**1. Introduction**

**line 112**: *then* should be *than*.

**line 124**: I would use *in the German Bight* (more places in the manuscript).

**2.2 The wave model WAM**

**line 169**: The first of the paragraph line speaks of version 4.5.4, but here it is 5.4.5 (and with a hyphen).

**2.4 Study period and data availability**

**line 210**: Figures 2 and 3 are not (hardly) on storm Xaver, and certainly do not show the minimum pressure.

**line 210, 211**: Use singular *high tide*: even though it occurs at different locations on the German Bight at different times, it is still the top of one tidal wave.

**line 214**: *at low tide* instead of *at low water time*? Or do you mean really something different?

**line 215ff**: something is wrong there.

**3.1 Altimeter data**

**line 253ff**: Bad sentence. The time series are of wave heights and wind speeds during storm Xaver.

**line 268**: *wave heights of 2 meter respectively*: something is missing there.

**line 268**: Note the inconsistency between *in-situ* here and *in situ* elsewhere.

**line 271ff**: Do not use the abbreviation *std* without at least one time explaining it in full.

**3.2 Altimeter-model comparison**

You compare the output of *wave models* to remotely sensed data. It would help if you tell here whether you mean 1-way versus 2-way coupled or also already North Sea versus German Bight.

**line 287**: In Section 3.1 you found that the altimeters underestimate the wind speed compared to the in-situ measurements. Then, to conclude that a better agreement of the models with the satellite data means a *skill improvement* seems a step too far. They are closer, but might suffer from the same bias.

**line 297**: *waves smaller than one metre*: Why 1 m? Notice also the inconsistent spelling of *metre*, in other places I see *meter*.

**line 307**: Something is wrong there

**line 314ff**: The modeled wave height is much smoother than the observations because the model does not resolve the small scales which you see in the observations. That has little to do with post processing.

**line 319**: You can not conclude that the peak is shifted northward: you are at the end of the satellite track (Why? The satellite should have data more North as well). The valid conclusion would be that you miss the observed peak just above 58°N, but that the field data suggest this might be outside the model area. But you can not rule out either that there is another peak there which you miss because of the broken satellite track.

**line 324ff**: this disagreement between model and observation does **not** indicate anything about the satellite algorithm. You might just remark that it confirms conclusions of Fenoglio-Marc or something like that.

**3.3 Validation against in situ measurements**

**line 342ff**: *The comparison … are exemplified* is grammatically incorrect, and you should probably formulate it completely different. Figure 7 gives the results and you are now going to compare them.

**line 352**: *due to the time shift in the wind data*: suggests that you explained this time shift somewhere earlier. But I can not find that.

**line 382**: *behaviour* is always singular

**line 396**: *reduced by 5%*: I read an *increase* of 2.5% in Table 3.

**4. Impact of the two-way coupling**

I am still not happy with the use itself of *bias* in this case. Bias indicates a deviation from a reference, but here you are just comparing two different model configuration of which neither should be considered as reference a-priori. The phrasing *average difference in wave height* is correct, but you should not call it *bias*.

Something similar applies to the use of *RMSE*. This stands for Root Mean Squared Error, but for that you also need a reference. In the original manuscript it was correctly called *root mean squared difference*.

**5. Summary and Outlook**

**line 342**: the use of *perform* is incorrect here.

**line 343**: I do not think that the coupling software *analyses* anything. It just couples two models.

**line 470**: *than from*? Probably *from* should go away.

**line 471**: *This study* is confusing: do you mean the current paper or Staneva et al.?

**line 475ff**: The way this is formulated suggests more than what is dealt with in this paper. The use of *largely* is probably not appropriate here, as is *nevertheless*.

**Figures**

**Figure 1b**

The name *Westerland* is still unreadable.

**Figure 2**

What exactly is the radial variable?

**Figure 3**

The first subfigure is quite different from the other two. Especially when the colours used are not the same: red for Saral/Altika in the map and blue in the time series. I would make a separate Figure 4 for the time series.

The caption of the new Figure 4 should more clearly indicate that it is the observations in station FINO-1 together with the Saral/Altika observation. The blue vertical lines in the time series should be removed: the square is the satellite observation.

**Figure 4 (old)**

No tick marks on the left axis of the right plot; the Y axis text is far too close to the left plot; the caption is a mess; *pattern* should be *panel*.

**Figure 6**

Figure 6a lacks the x-axis title *latitude*.

**Figure 7**

The yellow lines are still hardly visible. I actually meant in my earlier comments, that you should not use yellow for such lines at all.

**References**

I only looked up 3 or so in the references list, but of those 2 had incorrect years in the text. So, all references should be carefully checked.

---

## Referee Report (RR3)

The authors have addressed my comments and I am satisfied with the improvements.

Still, I have a few minor comments. The most important is that with the additions and alterations quite a few new language errors have been introduced. I will not list all of them, but the authors should carefully reread their manuscript, or have it corrected.

**2.4 Study period and data availability**

**line 200**: *wind-induced maximum*: as the surge is (mainly) due to wind, also the other maxima must be wind-induced. What the authors are explaining here is that there is a difference between a **locally** generated surge and a surge that is generated further away and propagates to the study area.

**line 230**: Should *DD* be *DDA*? Actually, in 254 there is a link between SAR and DDA. That should be made here already.

**3.2 Altimeter-model comparison**

**line 281**: You might want to avoid the abbreviation *SWH* altogether. It is introduced here, again in line 375, and very often you still use *significant wave height* which looks perfectly natural.

---

## Author Response (AR2)

**Authors response to the Reviewers' #1 comments**

Revised manuscript "Wave-atmospheric modelling, satellite and in situ observations in the Southern   North Sea: the impact of horizontal resolution and two-way coupling" by Kathrin Wahle et al.   presents interesting results of how two-way atmosphere-wave coupling improves the coastal wave   forecast. This is an important step in development of coupled models for short-term coastal   forecasting with high-resolution, especially as Authors have been able to show that the very high-   resolution coastal applications also benefit from the two-way coupling. The revision has improved   the manuscript, making it more focused and easier to read. The Authors have also well taken into   account the reviewers comments and suggestions.

Authors: Thank you for reviewing our manuscript again.

There are, however, still few places where further   clarification is needed (please see my detailed comments below).  Also, I suggest that some copy   editing is done to the text before publication. Some sentences are quite long and therefore not easy to follow and there is also some repetition.

Authors:  The revised manuscript has been crossed-checked again; additional copy editing has been also done. Following the reviewer's suggestion the long sentences have been split and we tried to avoid repetitions in the text.

Some detailed comments:

Section 1. Introduction: This section has improved a lot, and is now better structured and easier to follow. It is, however, quite long and has repetition. I think it would benefit of some copy-editing.

Authors:  We agree and the suggested revision has been done; the section is shortened and repetitions are removed.

Section 1, lines 58-59: alternative to fully-coupled ocean-atmosphere model? Would this model include also waves? And why should we have alternatives, shouldn't we aim for the fully coupled models.

Authors:  We agree and re-phrased the text in the introduction. Additionally we added in the discussion that: "Two-way coupling of wave and atmospheric models is an important component of a fully coupled ocean-atmosphere modelling system, as it resolves more adequately the interactions and exchanges in the atmospheric boundary layer."

Section 2.2, line 169: Should it be WAM4.5.4?

Authors:  Yes, thank you. This has been corrected.

Section 3.1, line 260: the situ-data → the in-situ data?

Authors:  We apologize for the mistake and corrected to "the *in-situ* data".

40    Section 3.2, first paragraph: The bias is calculated as measured minus modelled value and following  this, in text it is said that altimeter data underestimates the modelled values. Shouldn't this be said  the other way around? E.g. modelled values are overestimated compared to the satellite  measurements. In  most of the cases I'd assume the altimeter data to be more accurate than the  modelled data and it is said to be the dataset against which the model is verified.

45    Authors:  We agree and the suggested revision has been done.

      Section 3.2, first paragraph: If the Authors do not trust the Cryosat-swh, why is it used for validation in the first place. Wouldn't two altimeter datasets be enough for validation? Anyhow, the  explanation related to this in lines 287-297 is bit complicated to follow. Should the reader
50    disregard  the results from this comparison or interpret them with care?

      Authors:  We agree that the Cryosat-swh must be interpreted with care. This has been now better explained in the revised manuscript.  The text in lines 287-287 has been re-phrased following the reviewer's suggestion.

55    Section 3.2, lines 285-287: Quite a long sentence, could be split to two parts

      Authors: The sentence is split into two parts.

      Section 3.3, second paragraph: The reason behind the better behaviour of the high-resolution model   is probably mostly due to the better description of the bathymetry in the area, not the high-
60    resolution *per se.* It is implicitly mentioned by the Authors, but it could be stated more clear.

      Authors:  We agree that the reason behind the better behaviour of the high-resolution model is probably mostly due to the better description of the bathymetry in the area, not the high-resolution per se  and this is better stated in the revised manuscript.  The suggested revision has been done.

65

      Section 4, line 414: Why not simply say, that the fetch is too short for the waves to evolve.
      Authors: We re-phrased the sentences just explaining that "the fetch is too short for the waves to evolve".

70    Section 5, lines 446-449: Quite a long sentence, could be split to two parts.

      Authors: The sentence has been split to two parts.

      Section 5, lines 473-474: What is meant by potential uncertainties of shallow water in the wave model? Is this related to the description of bathymetry or to the wave model source terms related
75    to  shallow water physics?

Authors: The potential uncertainties of shallow water in the wave model are due to both: inaccurate description in the bathymetry as well as to the wave model source terms related to shallow water physics. This has been now clearer explained in the revised manuscript.

80    Tables 1-3: Table captions should explain what the red and green colouring means.

Authors: The suggested revision has been done.

Figure 3: Please explain the marked overflight also in the Figure caption.

Authors: The suggested revision has been done.

85    In the Figure and table captions there is a mixture of terms "wave height" and "significant wave height". Preferably "significant wave height" should be used in all of them.

Authors: The suggested revisions have been done.

**Authors response to the Reviewers' #2 comments**

90    The manuscript has certainly improved, but there is still work left.

Authors:

Thank you for reviewing our manuscript again. The manuscript has been again carefully checked and the suggested revisions have been done.

95    **Title**

I liked the old title better than the new one. The title should convey the essence of the paper, and for me that is the effect of a two-way coupling of an atmosphere and wave model on the representation of atmospheric and wave parameters in a shallow and complex coastal area. Explicitly mentioning both in-situ and satellite observations draws to much attention away from

100    that. And, moreover, the title is now too much a collection of loose terms without really connecting them.

Authors: In the revised manuscript the tile has been changed to the initial one.

**1. Introduction**

105    **line 112**: *then* should be *than*.

Authors: The suggested revision has been made.

**line 124**: I would use *in the German Bight* (more places in the manuscript).

Authors: We agree and changed the text accordingly.

110

**2.2 The wave model WAM**

**line 169**: The first of the paragraph line speaks of version 4.5.4, but here it is 5.4.5 (and with a hyphen).

Authors: We apologize for this incorrectness, which we now correct.

**2.4 Study period and data availability**

**line 210**: Figures 2 and 3 are not (hardly) on storm Xaver, and certainly do not show the minimum pressure.

Authors: We modified the text accordingly.

**line 210, 211**: Use singular *high tide*: even though it occurs at different locations on the German Bight at different times, it is still the top of one tidal wave.

Authors: The change has been made.

**line 214**: *at low tide* instead of *at low water time*? Or do you mean really something different?

Authors: The change has been made.

**line 215ff**: something is wrong there.

Authors: This sentence has been re-phrased.

**3.1 Altimeter data**

**line 253ff**: Bad sentence. The time series are of wave heights and wind speeds during storm Xaver.

Authors: The paragraph in this section was re-phrased.

**line 268**: *wave heights of 2 meter respectively*: something is missing there.

Authors: We apologize for the unclear statement; the sentence has been modified.

**line 268**: Note the inconsistency between *in-situ* here and *in situ* elsewhere.

Authors: The inconsistency has been corrected. We use now everywhere "*in-situ*".

**line 271ff**: Do not use the abbreviation *std* without at least one time explaining it in full.

Authors: The suggested revision has been made. We explained it in the text before using the abbreviation *STD*.

**3.2 Altimeter-model comparison**

You compare the output of *wave models* to remotely sensed data. It would help if you tell here whether you mean 1-way versus 2-way coupled or also already North Sea versus German Bight.

Authors: This was discussed in the introduction and then explained in details in Section 2.3.

**line 287**: In Section 3.1 you found that the altimeters underestimate the wind speed compared to the in-situ measurements. Then, to conclude that a better agreement of the models with the satellite data means a *skill improvement* seems a step too far. They are closer, but might suffer from the same bias.

Authors: We agree and modified this text accordingly. We also removed the statements about *"skill improvement…"* .

**line 297**: *waves smaller than one metre*: Why 1 m? Notice also the inconsistent spelling of *metre*, in other places I see *meter*. I do not understand the context smaller than one

Authors: We are sorry for the misspelling and changed "metre" to "meter". The sentence has been also changed. Explanation, including references has been added.

**line 307**: Something is wrong there

Authors: We removed the redundant part.

**line 314ff**: The modeled wave height is much smoother than the observations because the model does not resolve the small scales which you see in the observations. That has little to do with post processing.

Authors: We agree with this comment and the discussion on the comparisons (Fig. 5a,b in the revised manuscript) has been re-phrased accordingly.

**line 319**: You can not conclude that the peak is shifted northward: you are at the end of the satellite track (Why? The satellite should have data more North as well). The valid conclusion would be that you miss the observed peak just above 58°N, but that the field data suggest this might be outside the model area. But you can not rule out either that there is another peak there which you miss because of the broken satellite track.

Authors: We agree with the remark and this has been changed in the revised version, accordingly.

180 **line 324ff**: this disagreement between model and observation does **not** indicate anything about the satellite algorithm. You might just remark that it confirms conclusions of Fenoglio-Marc or something like that.

Authors: We agree with the comment and the discussion about the disagreement between model and observation has been modified.

185

**3.3 Validation against in situ measurements**

**line 342ff**: *The comparison ... are exemplified* is grammatically incorrect, and you should probably formulate it completely different. Figure 7 gives the results and you are now going to compare them.

190 Authors: The discussion on the comparisons (the results in Fig. 5a,b in the revised manuscript) has been re-phrased accordingly.

**line 352**: *due to the time shift in the wind data*: suggests that you explained this time shift somewhere earlier. But I can not find that.

195 Authors: In the revised manuscript the variability in the wind data and the shift of the simulated wind peak has been demonstrated and the time-shift explained before discussing the measured and modeled significant wave height comparisons.

**line 382**: *behaviour* is always singular

200 Authors: The suggested revision has been made.

**line 396**: *reduced by 5%*: I read an *increase* of 2.5% in Table 3.

Authors: We agree and apologize for the mistake. This has been corrected in the revised manuscript.

205

**4. Impact of the two-way coupling**

I am still not happy with the use itself of *bias* in this case. Bias indicates a deviation from a reference, but here you are just comparing two different model configuration of which neither should be considered as reference a-priori. The phrasing *average difference in wave height* is 210 correct, but you should not call it *bias*.

Authors: The suggested revisions have been made: instead of "bias" we used "average difference"

Something similar applies to the use of *RMSE*. This stands for Root Mean Squared Error, but 215 for that you also need a reference. In the original manuscript it was correctly called *root mean squared difference*.

Authors: Now instead of "Root Mean Squared Error "we used "root mean squared difference"

**5.   Summary and Outlook**

**line 342**: the use of *perform* is incorrect here.

Authors: This sentence has been re-phrased.

**line 343**: I do not think that the coupling software *analyses* anything. It just couples two models.

Authors: This sentence has been re-phrased.

**line 470**: *than from* ? Probably *from* should go away.

Authors: We apologize for the redundancy and removed "from" from the text.

**line 471**: *This study* is confusing: do you mean the current paper or Staneva et al.?

Authors: This sentence has been re-phrased. We meant the paper of Staneva et al.

**line 475ff**: The way this is formulated suggests more than what is dealt with in this paper. The use of *largely* is probably not appropriate here, as is *nevertheless*.

Authors: The suggested revision has been made.

**Figures**

**Figure 1b**

The name *Westerland* is still unreadable.

Authors: We agree and Figure 1b has been re-plotted, making the text readable.

**Figure 2**

What exactly is the radial variable?

Authors:  The wind rose diagram is described in the Figure 2 caption.

**Figure 3**

The first subfigure is quite different from the other two. Especially when the colours used are not the same: red for Saral/Altika in the map and blue in the time series. I would make a separate Figure 4 for the time series.

Authors: We split now the sub-plots in different figures (Figure 3 and 4 in the revised manuscript).

The caption of the new Figure 4 should more clearly indicate that it is the observations in station FINO-1 together with the Saral/Altika observation. The blue vertical lines in the time series should be removed: the square is the satellite observation.

255 Authors: We modified the capture of the new Figure 4 accordingly. Also the suggested revisions in the figures have been done.

**Figure 4 (old)**

No tick marks on the left axis of the right plot; the Y axis text is far too close to the left plot;
260 the caption is a mess; *pattern* should be *panel*.

Authors: The suggested revisions in the figures have been done. The quality of the figure has been improved.

**Figure 6**

265 Figure 6a lacks the x-axis title *latitude*.

Authors: The suggested revisions in the figures have been done.

**Figure 7**

The yellow lines are still hardly visible. I actually meant in my earlier comments, that you should
270 not use yellow for such lines at all.

Authors: In the revised manuscript we changed the colours of the lines.

**References**

I only looked up 3 or so in the references list, but of those 2 had incorrect years in the text. So,
275 all references should be carefully checked.

Authors: The references in the revised manuscript have been cross-checked.

[revised manuscript text omitted]

(a)               (b)

(c)               (d)

1050

[Figure]

*Figure 9̶10: (a,c) Average difference (̶b̶i̶a̶s̶)̶ and (b,d) rms difference (̶r̶m̶s̶e̶)̶ rms difference) of WAM modeled significant wave height (m, top panel) and COSMO modeled wind speed (m/s, bottom panel) when comparing one-way minus two-way coupled modeling results. The differences are calculated as a̶v̶e̶r̶a̶g̶e̶d̶averages over the whole three month period.*

1055

mean sea level pressure [Pa]

[Figure]

90000,0  93000,0  96000,0  99000,0  102000,0 105000,0

1060  *(a)*

msl pressure difference [Pa]

[Figure]

−50,00   −30,00   −10,00   10,00   30,00   50,00

*(b)*

*Figure 11: (a) COSMO pressure (Pa) at mean sea level height in the North Sea during storm 'Xaver'*

1065  *and (b) mean sea level pressure differences when comparing one-way minus two-way coupled modeling).*

[Figure]

*Figure 12: Time series of significant wave height (m, top), wind speed (m/s, middle) and wave age (bottom) from the two-way coupled German Bight setup at FINO-1 for (a) a rather calm period with young wind sea and (b) during the storm 'Xaver'). Red lines in the top and middle panel show the differences between the one-way and the two-way coupled models.*

[Figure]

*Figure 13: As Figure 12 but during the storm 'Xaver'*

1075

---

## Author Response (AR3)

**Authors response to the Reviewers' #1 comments**

The authors have addressed my comments and I am satisfied with the improvements.
Still, I have a few minor comments.
Authors: Thank you for reviewing our manuscript again.

The most important is that with the additions and alterations quite a few new language errors have been introduced. I will not list all of them, but the authors should carefully reread their manuscript, or have it corrected.
Authors: The revised manuscript has been carefully crossed-checked again; additional copy editing has been also done.

2.4 Study period and data availability
line 200: wind-induced maximum: as the surge is (mainly) due to wind, also the other maxima must be wind-induced. What the authors are explaining here is that there is a difference between a locally generated surge and a surge that is generated further away and propagates to the study area.
Authors: We agree and the suggested revision has been done.

line 230: Should DD be DDA? Actually, in 254 there is a link between SAR and DDA. That should be made here already.
Authors: The suggested revision has been done.

3.2 Altimeter-model comparison
line 281: You might want to avoid the abbreviation SWH altogether. It is introduced here, again in line 375, and very often you still use significant wave height which looks perfectly natural.
Authors: We agree and the suggested revision has been done.